# Sample-wise Constrained Learning via a Sequential Penalty Approach with Applications in Image Processing

## Abstract

In many learning tasks, certain requirements on the processing of individual data samples can arguably be formalized, in an intelligible way, as strict constraints of the underlying optimization problem, rather than by means of arbitrary penalties that often require heavy trial-and-error tuning. In this paper we show that, in these scenarios, learning can be carried out exploiting a sequential penalty method that allows to explicitly deal with constraints. For the proposed algorithm we show that, under classical assumptions and in the considered learning scenario, penalty subproblems can provably be solved in finite time with suitable guarantees of approximate stationarity in expectation; we also show that, when the latter condition is guaranteed for the subproblems, the overall sequential scheme possesses almost sure convergence properties to KKT-stationary points of (general) constrained problems. The results of experiments on sandbox and real-world image processing tasks show that the method is indeed viable to be used in practical deep learning scenarios.

## 1 Introduction

As the computational and expressive power of deep learning models keeps growing, leading to surprising breakthroughs in science and technology at a sustained pace (Jumper et al., 2021; Lam et al., 2023; Katz et al., 2024; Merchant et al., 2023), interest in the use of these techniques in new scenarios and for very specific applications is also rising. The requirements in some of these setups are really tailored and often come in the form of precise specifications for the outputs of the network. In mathematical terms, these requests would translate in the introduction of *constraints* within the learning task. A prime example of this situation occurs with image processing applications, where the network is required to apply some transformation within input images to achieve a main goal - e.g., insertion of a watermark (Zhu et al., 2018) or creation of adversarial samples (Xu et al., 2020) - while preserving visual perception quality.

To address these types of challenges, researchers often introduce some metric for measuring the quality of an output with respect to the requirement and then use it to define an additional loss function to be added to the main loss of the task at hand (Chakraborty et al., 2025; Chen et al., 2016; Dunion et al., 2023; Higgins et al., 2017; Kumar et al., 2018; Zhu et al., 2017; Magistri et al., 2024). In this way, behaviors of the network contrasting with the requirement are discouraged, and training can take into account the additional specification. This approach is particularly convenient for practitioners, as training can be performed as usual by SGD-type algorithms like Adam (Kingma & Ba, 2015), efficiently exploiting standard automatic differentiation libraries.

However, as also thoroughly underlined by Ramirez et al. (2025), the issue with the above strategy lies in the choice of the trade-off (hyper)parameter to be set within the overall loss, which is most often not intelligible by humans. The risk is therefore to select a value for the penalty term in the loss being either too low - resulting in the partial or even total neglect of the additional requirement by the resulting network - or too large - leading to the sacrifice of the performance with the main goal. In other words, there is a likely risk of either ignoring the constraints or sacrificing performance to satisfy it with unnecessary margin. For a proper calibration of the learning process a careful validation would thus be needed, with still possibly flawed results.

We thus align with the viewpoint expressed and supported in detail by Ramirez et al. (2025), a viewpoint actually noted some decades ago already by Platt & Barr (1987) and more recently affirmed by Lavado et al. (2023); Fioretto et al. (2020); Dener et al. (2020); Nandwani et al. (2019), and we argue that those requirements should actually be treated for what they essentially are: constraints of the learning optimization problem. In fact, the threshold value for the constraint can be intelligibly set by the user: with reference once again to the case of image processing, a human can straightforwardly identify the acceptability level for the perceptive distortion of the images. Over that threshold, output images should be rejected altogether; under that threshold, we should be fine and stop requiring further improvement of output visual quality. This type of path was for instance followed for imposing weights sparsity in the resulting network (Gallego-Posada et al., 2022), physics constraints (Dener et al., 2020; Hwang & Son, 2022), regularization (Lavado et al., 2023), class-balanced predictions (Sangalli et al., 2021), natural language semantic (Nandwani et al., 2019).

We are thus interested in studying constrained learning problems and, in particular, tasks where a clear constraint shall be satisfied by model output (or byproducts) for each data point:

$$\min_{w} \ \mathcal{L}(w) = \frac{1}{N} \sum_{i=1}^{N} \ell(w; x^i, y^i) \quad \text{s.t.} \ c(w; x^i) \leq B \quad \forall i, \tag{1}$$

where $\mathcal{L}$ represents the main loss function, dependent on the network tunable weights $w$, computed on a training set of $N$ samples, and $c$ is the constraint function, that shall take a value under the threshold $B$ for all samples in the dataset. Similarly to a loss function, the constraint $c$ is a function of the weights of the network and provides some metric related to the output of the network given an input vector. This scenario is for instance covered by Dener et al. (2020); Sangalli et al. (2021); Gnecco et al. (2014). We will not treat constraints that directly affect model structure - imposed, e.g., for regularization, model compression or physical consistency aims, instead analyzed by Hwang & Son (2022); Gallego-Posada et al. (2022).

We shall underline that, of course, constraints are set on training data: we will in any case have no guarantee that network outputs will also satisfy them for out-of-sample data. Yet, this issue is intrinsic with learning problems and would be equally troublesome if we solved, as often done in practice, the "penalized" problem

$$\min_{w} \mathcal{L}(w) + \lambda \sum_{i=1}^{N} c(w; x^i).$$

The focus of this work will be posed on the design of a suitable algorithmic framework for solving the specific class of problems (1) with the explicit management of the constraints. The optimization method we present within this work is a sequential penalty approach that makes variables updates via stochastic-gradient type steps. Sequential approaches, like penalty and augmented Lagrangian methods (ALMs) - see the books by Grippo & Sciandrone (2023, Ch. 21) or Birgin & Martínez (2014) for a detailed introduction - represent consolidated ways of tackling optimization problems with nonlinear constraints in fully deterministic scenarios. In recent years, settings have also been considered taking into account stochasticity, noise or finite-sum structure in the objective function (Zuo et al., 2025; Lavado et al., 2023; Krejić et al., 2025; Wang et al., 2017a) and possibly also in the constraints (Li et al., 2024). In the latter case, noisy access to constraints can correspond to the subsampling of a finite-sum type of constraints, where all subfunctions (i.e., data points) simultaneously contribute to the constraint value and the approximation does not allow to grasp exact information about the possible current violation. We need to point out that if we employ mini-batch sampling methods on problem (1) we get something inherently different: we in fact sample the set of constraints, getting the exact value for the constraints associated with selected data points. To tackle the specific setting of (1), ALM-type approaches have been proposed by Dener et al. (2020); Sangalli et al. (2021), but convergence and correctness aspects in the mini-batch optimization scenarios were not rigorously addressed.

For the sequential penalty algorithm presented in this work, introduced in Section 3 after a preliminary discussion in Section 2, we prove correctness and asymptotic convergence properties under classical assumptions (Section 4) and we show the results of computational experiments carried out on a simple preliminary test problem (Section 5.1); we then present in Section 5.2 the results of the application of the proposed methodology on a real task related to the watermarking of medical images.

The contributions of the manuscript can be summarized as follows:

- we formalize the sample-wise constrained scenario in deep learning, pointing out that relevant problems usually handled by regularization could naturally be cast to this formulation;

- we propose an algorithmic scheme, easily implementable by practitioners in standard software frameworks, to deal with the above class of training tasks;

- we revisit the SGD convergence result for the non-convex case by Vaswani et al. (2019) making explicit the worst case bound on the number of updates needed to drive the expected value of the gradient norm of a finite-sum objective below a given threshold;

- we provide the (almost sure) convergence analysis, to the best of our knowledge novel in the literature, for the sequential penalty method applied to a general class of problems in the case where the stationarity condition for the subproblems can be obtained only in expectation;

- we show the viability of the proposed methodology both on a sandbox problem and a real image processing task, offering some insights on the algorithm behavior.

## 2 Problem Statement

With the broadest possible perspective, the class of optimization problems we address in this paper is that of the form

$$\min_{x \in \mathbb{R}^n} \ f(x) \qquad \text{s.t. } g_i(x) \leq 0, \ i = 1, \ldots, m, \tag{2}$$

where $f : \mathbb{R}^n \to \mathbb{R}$, $g_i : \mathbb{R}^n \to \mathbb{R}$, $i = 1, \ldots, m$, are $L_f$-smooth and $L_{g_i}$-smooth functions respectively. We recall that a function $\varphi$ is $L$-smooth if it is continuously differentiable and the gradient $\nabla \varphi$ is Lipschitz-continuous with Lipschitz constant $L$. We also assume $f$ is lower bounded on $\mathbb{R}^n$ by some value $f^*$. We denote the feasible set by $S = \{x \in \mathbb{R}^n \mid g_i(x) \leq 0, \ i = 1, \ldots, m\}$.

For problems of this form, the well-known Karush-Kuhn-Tucker (KKT) conditions (see, e.g., Bertsekas (1999)) can be stated according to the next definition.

**Definition 1** (Karush–Kuhn–Tucker (KKT) conditions)**.** *Suppose that $f, g_1, \ldots, g_m$ are continuously differentiable functions. A point $x^*$ satisfies the* KKT conditions *if there exist multipliers $\lambda^* = (\lambda_1^*, \ldots, \lambda_m^*) \in \mathbb{R}^m$ such that*

$$\nabla f(x^*) + \sum_{i=1}^m \lambda_i^* \nabla g_i(x^*) = 0,$$

*and, for all $i = 1, \ldots, m$, $g_i(x^*) \leq 0$, $\lambda_i^* \geq 0$, and $\lambda_i^* g_i(x^*) = 0$.*

To turn KKTs into necessary optimality conditions, we need to assume some regularity condition, or constraint qualification (CQ), on the feasible set (Bertsekas, 1999). While more general and less restrictive CQs could be employed for the study, for the aims of this work we prefer not to overcomplicate the analysis and thus here we focus in particular on the following standard condition.

**Definition 2** (Linear Independence Constraint Qualification (LICQ))**.** *Let $x \in S$ and let $I(x)$ the set of active constraints at $x$, i.e., $I(x) = \{i \mid g_i(x) = 0\}$. We say that the* Linear Independence Constraint Qualification *(LICQ) for problem (2) holds at $x$ if gradients $\nabla g_i(x)$, $i \in I(x)$, are linearly independent.*

The LICQ can in fact be extended so that the definition can cover also infeasible points of the problem.

**Definition 3** (Extended Linear Independence Constraint Qualification (E-LICQ))**.** *Let $x \in \mathbb{R}^n$ and let $I_+(x)$ the set of active and violated constraints at $x$, i.e., $I_+(x) = \{i \mid g_i(x) \geq 0\}$. We say that the* Extended Linear Independence Constraint Qualification *(E-LICQ) for problem (2) holds at $x$ if gradients $\nabla g_i(x)$, $i \in I_+(x)$, are linearly independent.*

The above definition will be useful later in this work, when dealing with the convergence properties of the proposed algorithm. Of course, at a feasible point the E-LICQ collapses to the standard LICQ. We are now ready to state the necessary condition of optimality.

**Theorem 1.** *If $x^*$ is a local minimum of problem (2) and the LICQ holds at $x^*$, then $x^*$ satisfies the KKT conditions.*

The constrained deep learning problem (1) is a particular instance of problem (2). In fact, if we assume to have $m$ constraint functions $g_i$ to enforce for each training data point $j$, we end up with

$$\min_{x \in \mathbb{R}^n} \ f(x) = \frac{1}{N} \sum_{j=1}^{N} f_j(x) \quad \text{s.t.} \ g_{ij}(x) \le 0 \ \forall i, \ \forall j, \tag{3}$$

where $x$ denotes the weights of the network. While for most aspects related to the analysis of both the problem and the algorithm the particular structure of problem (3) does not need tailored adjustments and we could just focus on the general case (2), the sample-wise structure of both objective and constraints in the learning scenario will be central in the design of an actually employable method.

## 3 A Sequential Penalty Approach with Inexact Stochastic Solver

There is a vast and consolidated literature in the optimization field concerning algorithms to tackle problems of the form (2) and, in particular, focusing on sequential approaches like penalty and augmented Lagrangian methods (again, see the books by Grippo & Sciandrone, 2023; Birgin & Martínez, 2014). To keep the treatment simpler, here we will theoretically analyze the case of (quadratic) penalty approaches, that are based on the penalty function defined hereafter. Still, similar discussions could be made for different choices of penalty function, which might actually result more effective from a numerical standpoint.

**Definition 4.** *The quadratic penalty function associated with problem (2) is defined as*

$$P_\tau(x) = f(x) + \frac{\tau}{2} \sum_{i=1}^{m} \max\{0, g_i(x)\}^2.$$

In essence, the *sequential penalty method* generates a sequence $\{x^k\} \subseteq \mathbb{R}^n$ such that each $x^k$ is a (approximate) solution to the subproblem

$$\min_{x \in \mathbb{R}^n} P_{\tau_k}(x),$$

for increasingly large values of $\tau_k$. The rationale of the approach is that of optimizing the objective function with a penalty for constraints violations; as the weight of the penalty in the subproblem objective grows, solutions will be progressively encouraged to strictly satisfy the constraints. Convergence results for this scheme depend, intuitively, on how the subproblems are solved. Fortunately, it is not necessary to exactly solve each subproblem to global optimality. In standard setups, by "solving" we usually mean that an approximate stationary point for $P_{\tau_k}$ is found, meaning that $\|\nabla P_{\tau_k}(x^k)\| \le \epsilon_k$.

For convergence, we will then ask $\tau_k \to \infty$ and $\epsilon_k \to 0$, i.e., we get progressively more accurate as iterations go by and we work with increasingly penalized objectives. Under these and some other standard assumptions (see Grippo & Sciandrone, 2023, Ch. 21), the framework can be proven to enjoy the following property: if the produced sequence admits limit points, then all limit points are feasible for the original problem and satisfy KKT conditions. While not explicitly required in theory, minimization of $P_{\tau_k}$ shall start from $x^{k-1}$ for computational reasons.

In the interesting case of problems of the form (3), we therefore see that, if we divide the value of all constraints by $N$ (which does not alter the problem), the penalty function takes the form

$$P_\tau(x) = \frac{1}{N} \sum_{j=1}^{N} f_j(x) + \frac{\tau}{2N} \sum_{i=1}^{m} \sum_{j=1}^{N} \max\{0, g_{ij}(x)\}^2 = \frac{1}{N} \sum_{j=1}^{N} P_\tau^j(x),$$

where $P_\tau^j(x) = f_j(x) + \frac{\tau}{2} \sum_{i=1}^{m} \max\{0, g_{ij}(x)\}^2$. The penalty function is therefore a finite-sum function. Penalty subproblems can then be naturally handled and approximately solved by the usual SGD type methods employed for large-scale machine learning (Bottou et al., 2018). Of course, to proceed in this direction we

have to accept that approximate optimality results for subproblems will be only obtainable in expectation. In other words, we have to settle for a result of the type

$$\mathbb{E}[\|\nabla P_{\tau_k}(x^k)\|]] \leq \epsilon_k \tag{4}$$

for all $k$. The main challenges addressed in this work thus regard two key questions:

- Is it possible to devise a stopping condition for an SGD solver so that we can ensure equation 4 will be practically attained in a finite number of steps for all $k$?

- Can we prove asymptotic convergence properties for the sequence $\{x^k\}$ if equation 4 is satisfied for all $k$?

While we will focus on the specific case of problem (3) with finite-sum type penalty functions in the analysis of the inner optimization loop, the analysis for the outer algorithm will cover the more general case of problem (2) where the penalty subproblems are solved stochastically and equation 4 is enforced by any technique.

## 4 Convergence Analysis

In this Section, we present a theoretical convergence analysis demonstrating that, at least in ideal conditions and in its simplest form, we should expect the proposed algorithmic framework (whose formalization as a pseudocode can be found in Algorithm 1 of Appendix E) to be effective. To begin, we need to recall some important concepts. In what follows, $\mathbb{E}_i$ denotes the expected value w.r.t. the random variable $i$, which in turn denotes the randomly sampled term of the finite sum (i.e., the data point). We assume that sampling is conducted in such a way that $\mathbb{E}_i[\nabla \phi_j(x)] = \nabla \phi(x)$, i.e., sampled gradient is an unbiased estimate of full gradient $\nabla \phi(x)$. We introduce now an assumption commonly made in the analysis of SGD algorithms, which regards the stochastic gradients and that holds true for relevant neural architectures that operate in the overparametrized regime (Mishkin, 2020; Liu et al., 2022).

**Definition 5** (Schmidt & Roux 2013). *A finite-sum function $\phi(x) = \sum_{j=1}^{N} \phi_j(x)$ satisfies the Strong Growth Condition (SGC) if there exists $\rho > 0$ such that, for any point $x \in \mathbb{R}^n$, $\mathbb{E}_i[\|\nabla \phi_i(x)\|^2] \leq \rho \|\nabla \phi(x)\|^2$.*

We then have to recall a series of concepts and standard results from probability theory (see, e.g., Durrett, 2019, for reference) that will be needed to characterize and analyze the behavior of our stochastic procedure. We start with a classical concept of convergence in a non-deterministic scenario.

**Definition 6** (Convergence in probability). *Let $\{Y_k\}$ be an infinite sequence of random variables. We say that $Y_k$ converges in probability to $X$, written $Y_k \xrightarrow{P} X$, if for every $\varepsilon > 0$ we have*

$$\lim_{k \to \infty} \mathbb{P}(|Y_k - X| > \varepsilon) = 0.$$

Another standard convergence concept, strictly stronger than convergence in probability, is almost sure convergence.

**Definition 7** (Convergence almost surely). *Let $\{Y_k\}$ be an infinite sequence of random variables. We say that $Y_k$ converges almost surely to $X$, written $Y_k \xrightarrow{a.s.} X$, if $\mathbb{P}(\lim_{k \to \infty} Y_k = X) = 1$.*

As aforementioned, almost sure convergence implies convergence in probability. In general, the converse is not necessarily true. However, the following result can be stated.

**Lemma 1** (Durrett 2019, Th. 2.3.2). *A sequence $\{Y_k\}$ of random variables converges to $X$ in probability iff for every subsequence $\{Y_k\}_K$, $K \subseteq \{0, 1, \ldots\}$, there exists a further subsequence $\{Y_k\}_{K_1}$, $K_1 \subseteq K$, that converges to $X$ almost surely.*

In other words, while convergence in probability does not generally imply almost sure convergence, it does at least implies convergence of some subsequences. We finally conclude the preliminary discussion with a standard inequality.

**Lemma 2** (Markov's inequality)**.** *Let $X$ be a non-negative random variable, and let $a > 0$. Then $\mathbb{P}(X \geq a) \leq \frac{\mathbb{E}[X]}{a}$.*

We can now turn to the analysis of the algorithm.

### 4.1   Finite termination of the inner solver

Before delving into the main result of this section, we first need to understand the regularity properties of function $P_{\tau_k}(x)$: we shall indeed note that a general $P_\tau(x)$, associated with any problem of the form (2), is $L$-smooth in a compact set.

**Lemma 3.** *Let $C \subseteq \mathbb{R}^n$ be a convex compact set. The penalty function $P_\tau$ associated with problem (2) is $L_{\tau,C}$-smooth with $L_{\tau,C} = L_f + \tau\big(\sum_{i=1}^m M_{i1}^2 + M_{i2}L_g\big)$, where*

$$M_{i1} := \sup_{x \in C} \|\nabla g_i(x)\|, \qquad M_{i2} := \sup_{x \in C} \max(0, g_i(x)).$$

*Proof.* The proof is postponed to Appendix A. □

We can now analyze the convergence of SGD on penalty subproblems of the form

$$\min_x \ P_{\tau_k}(x) = \frac{1}{N}\sum_{j=1}^N P_{\tau_k}^j(x). \tag{5}$$

At this point, we are able to state the next result, which specializes well-known properties to provide a precise complexity bound linking the expected value of the norm of penalty function gradient and the number of SGD iterations.

**Theorem 2.** *Let $\{z^t\}$ be the sequence produced by SGD, with a constant stepsize $\eta = \frac{1}{\rho_{\tau_k} L_{\tau_k,C}}$ applied to problem (5), i.e., by updates of the form $z^{t+1} = z^t - \eta \nabla P_{\tau_k}^{i_t}(z^t)$, assuming that $P_{\tau_k}$ satisfies the SGC property with an SGC constant $\rho_{\tau_k}$. Further assume that there exists a convex compact set $C \subseteq \mathbb{R}^n$ such that $\{z^t\} \subseteq C$ and that, at each iteration $t$, the algorithm outputs a solution $\hat{x}^t$ uniformly drawn from $\{z^0, \ldots, z^{t-1}\}$, i.e., $\hat{x}^t \sim \mathcal{U}[z^0, \ldots, z^{t-1}]$. Then, for any $\epsilon_k > 0$, we have*

$$\mathbb{E}[\|\nabla P_{\tau_k}(\hat{x}^t)\|] \leq \epsilon_k$$

*for all $t \geq T_k$, with $T_k = \frac{2\rho_{\tau_k} L_{\tau_k,C}(P_{\tau_k}(z^0) - P_{\tau_k}^*)}{\epsilon_k^2}$.*

*Proof.* The proof is postponed to Appendix A. □

The result from Theorem 2 guarantees us that if SGD is run long enough on each subproblem, we eventually get a solution $x^k$ that provably satisfies equation 4. We also get an estimate of the number of iterations required to make the approximate stationarity property hold, which could thus be used for a practical stopping condition. However, this condition is quite impractical - also taking into account that some of the constants defining $T_k$ will in general be not known. Still, it is valuable from the theoretical perspective. An interesting insight, on the other hand, is that at each outer iteration, i.e., for larger values of $\tau$ and smaller values for $\epsilon$, we would be in principle asked to run SGD longer on the subproblem. Still, if $\tau_k$ grows slow enough and the optimization of $P_{\tau_k}$ starts from the approximate minimizer of $P_{\tau_{k-1}}$, then the gap $P_{\tau_k}(z^0) - P_{\tau_k}^*$ will often be reasonably small, mitigating this issue.

### 4.2   Outer Loop Convergence

We can now turn to the convergence analysis for the outer loop. As anticipated, the results here are valid for any problem of the form (2), provided we have access to an inner solver that provably satisfies equation 4 in finite time for each $k$.

The convergence result is reported in the following theorem. To the best of our knowledge, an almost sure convergence result of this kind is novel in the literature of penalty methods and thus represents the main theoretical contribution of this manuscript.

**Theorem 3.** *Consider problem (2) and let $P_\tau$ be the associated penalty function. Assume $C \subseteq \mathbb{R}^n$ is a compact set and $\{x^k\} \subseteq C$ is such that*

$$\mathbb{E}\big[\|\nabla P_{\tau_k}(x^k)\|\big] \le \epsilon_k,$$

*for two sequences $\{\tau_k\}$ and $\{\epsilon_k\}$ such that $\tau_k \to \infty$ and $\epsilon_k \to 0$. Then $\|\nabla P_{\tau_k}(x^k)\| \xrightarrow{P} 0$ and there exists a subsequence of indices $\{k_j\}$ such that $\|\nabla P_{\tau_{k_j}}(x^{k_j})\| \xrightarrow{a.s.} 0$. Moreover, almost surely there exists a limit point $\bar{x}$ of $\{x^k\}$ such that, if it satisfies the E-LICQ, then it is a feasible solution for problem (2), i.e., $\bar{x} \in S$, and it is a KKT point for the original problem.*

*Proof.* The proof is postponed to Appendix B. $\qquad\square$

## 5   Computational Experiments

The computational viability of the sequential penalty approach discussed in this work was evaluated taking into account two image processing applications, in which the presence of a strict requirement on the model behavior can be naturally expressed introducing a set of constraints on the training data, resulting in instances of the form equation 2. The sequential penalty method is compared to the baseline, usual approach of considering an additional term in the loss with fixed weight, that accounts for the additional requisite. The code is provided in the supplementary material.

In the experiments reported in this section we consider the sequential approach equipped with an absolute (linear) penalty function, rather than the quadratic one. This choice was motivated by the results of preliminary experiments on the first test problem, detailed in Appendix C, where we observed that the quadratic penalty exhibits a less stable behavior in practice. While feasibility is satisfied in the limit, to achieve small violations it is hence often necessary to drive the penalty parameter to very large values, which in turn make the optimization of the penalty function numerically troublesome (see, for instance, the discussion by (Bertsekas, 1999, Ch. 4.2)). In addition, even when the algorithm is terminated at very large values of $\tau$, quadratic penalties tend to induce very small, yet nonzero violations across all data points; this issue is related to gradients of the penalty term that tend to vanish by definition when violations are small ($\nabla(\max\{0, g(x)\}^2) = 2\max\{0, g(x)\}\nabla g(x)$ gets small when $g(x)$ is positive but close to zero). On the contrary, the sparsifying effect of the $\ell_1$ norm makes the error exactly zero for a large bunch of data points when $\tau$ gets sufficiently large, so that we finally retrieve a solution that is truly satisfying the constraints for most points in the training set. We refer the reader to the discussion by (Nocedal & Wright, 2006, Ch. 1.7) for insights on the effects that different norms induce on the residuals distribution in minimum error problems.

While we recognize that the setting considered in the experiments is not directly covered by the analysis from Section 4 - both in terms of assumptions and choice of the penalty function - the aim of this computational study is to provide a first feedback about the actual potential of the proposed methodology. The specialized analysis of the algorithm exactly used in the experiments would arguably shift the focus and main scope of this work, and we thus defer it to future research.

### 5.1   Preliminary algorithm study

The first experiment consists in a classification task on the MNIST database of handwritten digits using a multi-layer perceptron, where we additionally impose that the hidden representation can be used to reconstruct the original image in a trainable decoder-like branch of our network, so that the reconstructed image and the original image are close enough in terms of the mean squared error (MSE) on pixels values. More precisely, the $28 \times 28$ input image $I_j$ goes through two hidden layers of 256 and 20 ReLU-activated units respectively, producing the encoded 20-dimensional sample $v_j$. Then, $v_j$ is processed by two branches of

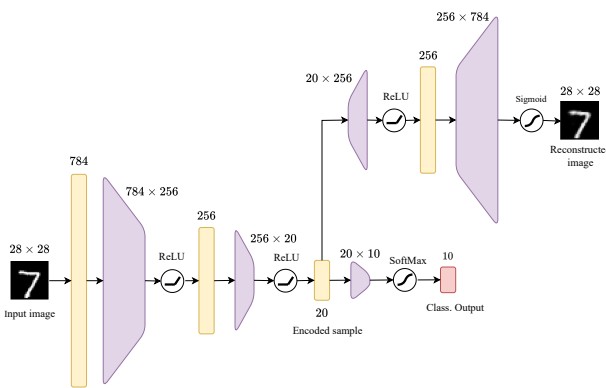

Figure 1: Network architecture for the toy problem: input images go through two fully connected layers mapping to a 20-dimensional encoding, then forwarded to distinct branches to get class predictions and the reconstructed images.

the model to produce the classification prediction $\hat{y}_j$, using a fully connected layer of 10 units with softmax activation, and the reconstructed image $\hat{I}_j$ through two fully connected layers of 256 and 784 units using ReLU and sigmoid activation respectively. The network architecture can be visualized in Figure 1.

In our setting we would like to train the model to obtain the best possible classification performances, measured with the cross entropy loss (CE), while having a reconstruction error below a certain threshold. Therefore, the problem can be formalized as follows

$$\min_{x \in \mathbb{R}^n} \frac{1}{N} \sum_{j=1}^{N} \ell_{\text{CE}}(y_j, \hat{y}_j) \quad \text{s.t.} \quad \ell_{\text{MSE}}(I_j, \hat{I}_j) \leq \theta \quad \forall j = 1, \ldots, N,$$

where $\ell_{CE}$ is the cross entropy loss for classification, $\ell_{\text{MSE}}$ is the pixel-wise mean squared error on image reconstruction and $\theta > 0$ is the tolerated reconstruction error. For the sequential penalty approach, which was implemented following the pseudocode provided in Algorithm 2 of Appendix E, we choose to increase $\tau$ at the end of each epoch, using the update rule $\tau_{k+1} = \gamma \tau_k$, with $\gamma > 1$ and $\tau_0 > 0$.

The sequential penalty scheme is compared to the classical fixed penalization approach, where the training problem is formalized as

$$\min_{x \in \mathbb{R}^n} \frac{1}{N} \sum_{j=1}^{N} \ell_{\text{CE}}(y_j, \hat{y}_j) + \lambda \ell_{\text{MSE}}(I_j, \hat{I}_j),$$

for some $\lambda > 0$. To get a reference for the highest performance achievable for classification with this architecture, we will also be considering the case of $\lambda = 0$, where the decoder is ignored during the training procedure.

For all approaches, optimization steps are always carried out by Adam (Kingma & Ba, 2015) with learning rate set to 0.001, weight decay of 0.001, $\beta_1 = 0.9$, $\beta_2 = 0.999$ and a batch size of 128. Note that, differently from the choice of the penalty function, the choice of Adam is somewhat theoretically justified by the results by Zhang et al. (2022). The encoding layers and the classification branch are warm-started providing a network pretrained for 5 epochs only considering the classification loss. Each considered method is then trained for 250 epochs. We set the maximal reconstruction threshold $\theta$ to 0.01. We run the experiment with three different seeds to avoid fluctuations in the results.

In Tables 1 and 2 we report the train and the test performances of the models trained with the sequential penalty method and with the fixed regularization (possibly with $\lambda = 0$). Together with the loss values for classification and reconstruction error, we report the classification accuracy, the average violation of the constraints $\ell_{\text{MSE}}(I_j, \hat{I}_j) \leq \theta$, and the percentage of satisfied constraints. For the latter two metrics, we

Table 1: Train performances of the sequential penalty method, of the fixed regularization method and, as a reference, of the model trained ignoring the reconstruction loss in the constrained MNIST reconstruction and classification problem. Average values over three runs are reported, together with standard deviation.

| [TRAIN] | $\ell_{\mathrm{CE}}$ ($\times 10^{-2}$) | Accuracy (%) | $\ell_{\mathrm{MSE}}$ ($\times 10^{-3}$) | Violation ($\times 10^{-4}$) | Satisfied (%) |
|---|---|---|---|---|---|
| Classification Only | $3.650 \pm 0.241$ | $99.19 \pm 0.05$ | $112.003 \pm 0.000$ | $102.003 \pm 0.000$ | $0.00 \pm 0.00$ |
| Fixed ($\lambda = 10$) | $4.149 \pm 0.350$ | $98.88 \pm 0.19$ | $17.914 \pm 0.089$ | $8.407 \pm 0.087$ | $16.78 \pm 0.08$ |
| Fixed ($\lambda = 100$) | $6.073 \pm 0.070$ | $98.28 \pm 0.03$ | $8.592 \pm 0.004$ | $1.457 \pm 0.006$ | $68.53 \pm 0.04$ |
| Fixed ($\lambda = 1000$) | $14.763 \pm 0.282$ | $95.63 \pm 0.10$ | $6.351 \pm 0.037$ | $0.538 \pm 0.010$ | $85.21 \pm 0.24$ |
| Penalty ($\tau_0 = 100, \gamma = 1.005$) | $7.075 \pm 0.192$ | $97.95 \pm 0.09$ | $9.000 \pm 0.043$ | $1.024 \pm 0.015$ | $69.28 \pm 0.09$ |
| Penalty ($\tau_0 = 100, \gamma = 1.01$) | $9.934 \pm 0.038$ | $97.09 \pm 0.05$ | $8.206 \pm 0.059$ | $0.600 \pm 0.020$ | $77.68 \pm 0.31$ |
| Penalty ($\tau_0 = 100, \gamma = 1.02$) | $16.180 \pm 0.319$ | $95.21 \pm 0.06$ | $7.704 \pm 0.010$ | $0.303 \pm 0.003$ | $84.22 \pm 0.10$ |
| Penalty ($\tau_0 = 10, \gamma = 1.01$) | $5.631 \pm 0.120$ | $98.42 \pm 0.06$ | $10.865 \pm 0.031$ | $2.200 \pm 0.012$ | $51.93 \pm 0.39$ |
| Penalty ($\tau_0 = 50, \gamma = 1.01$) | $8.404 \pm 0.048$ | $97.54 \pm 0.03$ | $8.650 \pm 0.030$ | $0.829 \pm 0.004$ | $72.73 \pm 0.03$ |
| Penalty ($\tau_0 = 200, \gamma = 1.01$) | $12.070 \pm 0.228$ | $96.45 \pm 0.06$ | $7.940 \pm 0.028$ | $0.461 \pm 0.013$ | $80.67 \pm 0.03$ |
| Penalty ($\tau_0 = 1000, \gamma = 1.01$) | $18.236 \pm 0.256$ | $94.59 \pm 0.06$ | $7.699 \pm 0.060$ | $0.282 \pm 0.015$ | $84.43 \pm 0.78$ |

Table 2: Test performances of the sequential penalty method, of the fixed regularization method and, as a reference, of the model trained ignoring the reconstruction loss in the constrained MNIST reconstruction and classification problem. Average values over three runs are reported, together with the standard deviation.

| [TEST] | $\ell_{\mathrm{CE}}$ ($\times 10^{-2}$) | Accuracy (%) | $\ell_{\mathrm{MSE}}$ ($\times 10^{-3}$) | Violation ($\times 10^{-4}$) | Satisfied (%) |
|---|---|---|---|---|---|
| Classification Only | $6.769 \pm 0.242$ | $97.97 \pm 0.10$ | $113.958 \pm 0.000$ | $103.958 \pm 0.000$ | $0.00 \pm 0.00$ |
| Fixed ($\lambda = 10$) | $7.367 \pm 0.313$ | $97.68 \pm 0.15$ | $17.406 \pm 0.084$ | $7.934 \pm 0.082$ | $17.95 \pm 0.01$ |
| Fixed ($\lambda = 100$) | $9.516 \pm 0.054$ | $97.08 \pm 0.10$ | $8.553 \pm 0.006$ | $1.454 \pm 0.011$ | $68.59 \pm 0.14$ |
| Fixed ($\lambda = 1000$) | $14.962 \pm 0.060$ | $95.52 \pm 0.05$ | $6.688 \pm 0.049$ | $0.706 \pm 0.013$ | $82.54 \pm 0.25$ |
| Penalty ($\tau_0 = 100, \gamma = 1.005$) | $9.806 \pm 0.469$ | $96.98 \pm 0.08$ | $9.097 \pm 0.051$ | $1.210 \pm 0.025$ | $68.07 \pm 0.51$ |
| Penalty ($\tau_0 = 100, \gamma = 1.01$) | $11.765 \pm 0.435$ | $96.53 \pm 0.03$ | $8.463 \pm 0.053$ | $0.923 \pm 0.021$ | $73.78 \pm 0.14$ |
| Penalty ($\tau_0 = 100, \gamma = 1.02$) | $15.964 \pm 0.570$ | $95.34 \pm 0.18$ | $8.318 \pm 0.022$ | $0.898 \pm 0.014$ | $74.60 \pm 0.40$ |
| Penalty ($\tau_0 = 10, \gamma = 1.01$) | $8.878 \pm 0.346$ | $97.33 \pm 0.13$ | $10.748 \pm 0.022$ | $2.181 \pm 0.003$ | $53.38 \pm 0.16$ |
| Penalty ($\tau_0 = 50, \gamma = 1.01$) | $10.794 \pm 0.162$ | $96.72 \pm 0.05$ | $8.813 \pm 0.029$ | $1.073 \pm 0.007$ | $70.30 \pm 0.26$ |
| Penalty ($\tau_0 = 200, \gamma = 1.01$) | $12.816 \pm 0.298$ | $96.17 \pm 0.09$ | $8.317 \pm 0.025$ | $0.874 \pm 0.004$ | $74.91 \pm 0.22$ |
| Penalty ($\tau_0 = 1000, \gamma = 1.01$) | $18.123 \pm 0.296$ | $94.85 \pm 0.08$ | $8.345 \pm 0.090$ | $0.895 \pm 0.034$ | $74.30 \pm 1.01$ |

might remark that even by the result from Theorem 3, feasibility is guaranteed to hold almost surely only in the limit, so we shall not be surprised that constraints are not all exactly satisfied when training stops in finite time. Multiple choices of $\tau_0$ and $\gamma$ in the sequential penalty method allowed to get good classification accuracy together with a large number of satisfied constraints both in the train and test set. Exceptions occur only for extreme choices.

The fixed penalty approach on the other hand appears more delicate to tune, as changing the order of magnitude for $\lambda$ massively impacts the behavior of the learned model: for $\lambda = 10$ the reconstruction error requirement is almost ignored, whereas for $\lambda = 1000$ the classification performances are heavily sacrificed to obtain good reconstruction.

In Figure 2 we report the distribution of reconstruction errors $\ell_{\mathrm{MSE}}(I_j, \hat{I}_j)$ across training and test data for the best performing setups of the sequential and fixed penalty approaches. We observe for our proposed method two interesting insights: a) a smaller tail of large violations is obtained by asking to satisfy constraints instead of penalizing the objective; b) the model does not unnecessarily push the reconstruction quality too far beyond the required threshold and, in particular, towards zero - avoiding needless accuracy drops. We report in Figure 8 of Appendix C the distribution of reconstruction errors obtained considering different parameters for the sequential penalty method.

In Figure 3 we report the accuracy and the percentage of satisfied constraints in both the train and test set during the training with the sequential penalty method, with the fixed regularization approach and the classification-only model. Compared to the classical approach, the sequential method adapts to obtain in

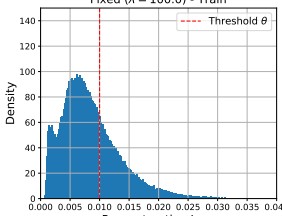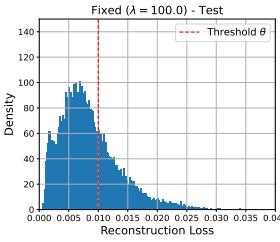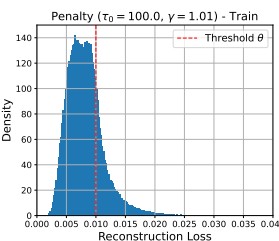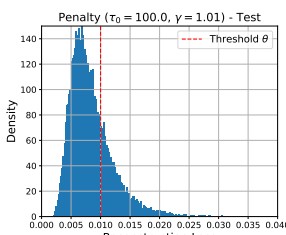

Figure 2: Train and test densities of the reconstruction loss for the sequential penalty method and for the fixed regularization method in the constrained MNIST reconstruction and classification problem. Density distributions are obtained combining the reconstruction losses of three independent runs.

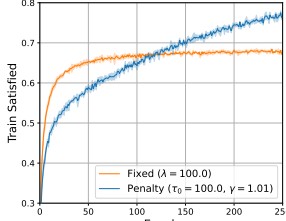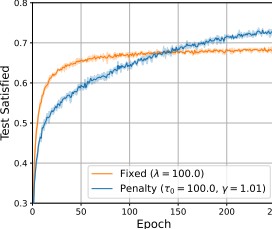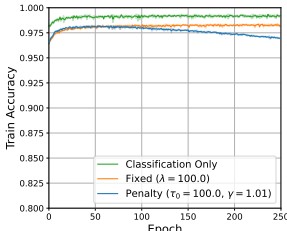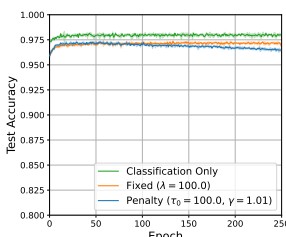

Figure 3: Train and test accuracy and percentage of satisfied constraints during the training with the sequential penalty method, the fixed regularization approach, and, as a baseline, only considering the classification loss in the constrained MNIST reconstruction and classification problem. Average over three runs is reported.

the end a high number of satisfied constraints as the penalty term increases, at the inevitable cost of a yet very limited decrease in classification accuracy.

## 5.2   A case study: Medical Image Watermarking

A second, more significant experiment focused on the watermarking of medical images. Digital watermarking refers to the process of embedding hidden information into multimedia content, such as images, through small and typically imperceptible modifications (Podilchuk & Delp, 2002). This technique has been successfully applied in a variety of real-world tasks, including copyright protection, traitor tracing, and metadata embedding. Whereas earlier watermarking methods relied on model-based algorithms grounded in signal processing theory, contemporary approaches frequently employ neural networks trained to encode and extract information while maintaining the perceptual quality of the underlying content.

This paradigm aligns naturally with our framework, as the training of such neural networks typically involves the joint optimization of two loss terms: one minimizing retrieval error for the embedded data, and the other maximizing the perceptual fidelity of the watermarked content. These objectives are inherently competing, since improving retrieval performance generally requires embedding more information, which in turn causes larger (and potentially perceptible) modifications to the original image. Balancing these goals is often challenging, as it usually depends on tuning a hyperparameter that lacks a clear semantic interpretation. This issue is especially problematic in domains such as medical imaging, where diagnostic images must satisfy stringent quality standards to ensure that their diagnostic utility is not compromised.

To evaluate our approach in this context, we constructed an experimental scenario designed to highlight its advantages. In particular, we adapted HiDDeN (Zhu et al., 2018), a well-established neural-network-based watermarking scheme, to operate within our proposed framework. This scheme, as in Figure 4, consists of an encoder that embeds a secret message $M_{in}$ into a cover image $I_{co}$ to produce a watermarked image $I_{en}$;

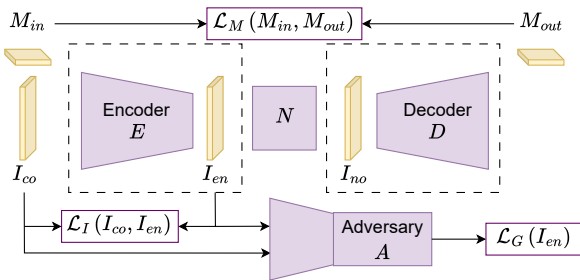

Figure 4: HiDDeN overview. Our proposed solution replaces $\mathcal{L}_I$ with a PSNR-based constraint.

a noise layer $\mathcal{N}(\cdot)$ that generates a possibly degraded version of $I_{en}$, $I_{no} = \mathcal{N}(I_{en})$; a decoder that retrieves a (potentially corrupted) message $M_{out}$ from $I_{no}$; and an adversarial discriminator trained to distinguish between $I_{en}$ and $I_{co}$. In its original formulation, the loss function for each sample

$$\mathcal{L}_M(M_{in}, M_{out}) + \lambda_I \mathcal{L}_I(I_{co}, I_{en}) + \lambda_G \mathcal{L}_G(I_{en})$$

balances three terms: the message distortion loss $\mathcal{L}_M$, which measures the reconstruction error of the embedded message; the image distortion loss $\mathcal{L}_I$, which quantifies the degradation introduced during watermarking; and the adversarial loss $\mathcal{L}_G$. During training, the losses $\mathcal{L}_I$ and $\mathcal{L}_G$ encourage the network to minimize perceptual distortion, whereas $\mathcal{L}_M$ promotes robust message embedding, implicitly pushing the network to introduce larger modifications to the image. The trade-off between robustness and distortion is governed by the hyperparameters $\lambda_I$ and $\lambda_G$, which, however, lack clear semantic interpretation.

In our experiment, we replace the image distortion term $\mathcal{L}_I(I_{co}, I_{en})$ with a constraint based on one of the most widely used metrics for assessing watermark imperceptibility: the Peak Signal-to-Noise Ratio (PSNR). PSNR quantifies the ratio between the maximum attainable power of a signal and the power of the noise that degrades its representation. Owing to the typically large dynamic range of image signals, PSNR is expressed in decibels. For two images $X$ and $Y$, PSNR is defined as $\text{PSNR}(X, Y) = 10 \log_{10}(\text{MAX}^2/\text{MSE}(X, Y))$, where $\text{MAX}^2$ denotes the maximum possible pixel value of images $X$ and $Y$, and $\text{MSE}(X, Y)$ is the mean squared error between them. Given that a high PSNR between the host image and the encoded image indicates strong perceptual similarity, it provides a semantically meaningful threshold: specifying a minimum PSNR value directly corresponds to enforcing a maximum allowable distortion level, making the constraint interpretable in terms of perceptual image quality.

Formally, the resulting training problem to be solved via sequential penalty is

$$\min_w \frac{1}{N} \sum_{j=1}^N \mathcal{L}_M\left(M_{in}^j, M_{out}^j(w)\right) + \lambda_G \mathcal{L}_G\left(I_{en}^j(w)\right)$$

$$\text{s.t. } \text{PSNR}\left(I_{co}^j, I_{en}^j(w)\right) \geq C, \qquad \forall j = 1, \ldots, N,$$

where $C$ denotes the required PSNR threshold. The model is encouraged to consistently produce watermarked images with PSNR values exceeding $C$.

To evaluate the effectiveness of our proposed strategy, we trained four different models employing the ChestX-ray8 dataset (Wang et al., 2017b). The dataset comprises 112120 frontal-view X-ray images of 30805 patients, with a native resolution $1024 \times 1024$. Each image is annotated with multiple labels including fourteen common thoracic pathologies: Atelectasis, Consolidation, Infiltration, Pneumothorax, Edema, Emphysema, Fibrosis, Effusion, Pneumonia, Pleural Thickening, Cardiomegaly, Nodule, Mass and Hernia. All models were trained on a custom 70/15/15 train/validation/test splits for 200 epochs using Adam with learning rate $10^{-4}$ and batch size 32. Images were resized to $3 \times 224 \times 224$ to meet the input requirements of the HiDDeN architecture. During training, the noise layer $\mathcal{N}$ was set to the identity function, and the watermark message length was fixed to $L = 200$ bits. We adopt the following notation to distinguish the experimental settings:

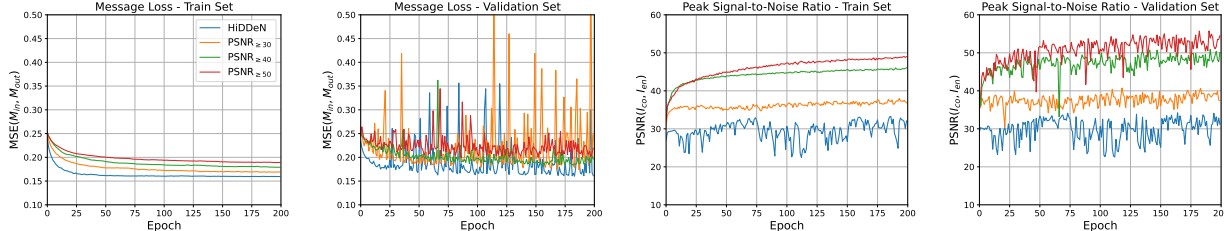

Figure 5: Training and validation curves for message loss $\mathcal{L}_M\left(M_{in}, M_{out}\right)$ and PSNR $\left(I_{co}, I_{en}\right)$ for all model configurations.

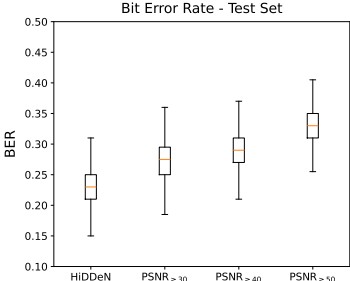

Figure 6: Bit Error Rate (BER) of the trained models on the test set.

- HiDDeN: baseline model, trained with weight factors $\lambda_I = 0.7$ and $\lambda_G = 10^{-3}$;

- PSNR$_{\geq C}$: proposed model with penalty coefficient $\tau$ increased by 10% every 10 epochs and $\lambda_G = 10^{-3}$; we consider the quality threshold values $C = 30, 40, 50$.

Figure 5 reports the training and validation curves of message loss $\mathcal{L}_M\left(M_{in}, M_{out}\right)$ and PSNR $\left(I_{co}, I_{en}\right)$ for all models. Compared to HiDDeN, the constrained models reach higher values on $\mathcal{L}_M\left(M_{in}, M_{out}\right)$, as increased imperceptibility necessarily trades off robustness. Overall, the message loss curves are similar in both shape and values, indicating that the models have comparable performances. With respect to PSNR $\left(I_{co}, I_{en}\right)$, HiDDeN, shows moderate variance and remains centered around a PSNR of approximately 30 on both training and validation. In contrast, models trained with penalty loss show a consistent pattern: (average) PSNR increases rapidly, finally converging to a plateau placed above the imposed threshold. This apparent overshoot is due to the fact that each sample is actually forced to meet the PSNR constraint, as it is confirmed by the probability density plots of PSNR values at the end of training that we report in Figure 9 of Appendix D. These plots demonstrate that the proposed incremental penalty strategy effectively enforces the desired image quality constraint, though at the expected cost of a higher bit error rate (BER) as the PSNR threshold increases, as shown in Figure 6. Pixel-wise comparisons between the original and watermarked images in Figure 7 reveal that watermark traces remain imperceptible, with visibility further decreasing at higher PSNR thresholds.

As a final experiment, we assessed the impact of watermarking on downstream pathology classification performance, employing a DenseNet-121 (Huang et al., 2017) multi-label classifier finetuned on data by Wang et al. (2017b), with the same data split used in the previous comparison among models. Specifically, Table 3 reports the AUROC scores of the classifier for each class on the original test set (denoted as *Base Classifier*), as well as the change in AUROC observed when the classifier is applied to watermarked images generated by different model configurations, relative to the original images. Interestingly, images watermarked with the HiDDeN model exhibited a significant drop in AUROC across all classes, whereas those generated with the PSNR-constrained variants showed minimal degradation for some classes in PSNR$_{\geq 30}$ and no observable

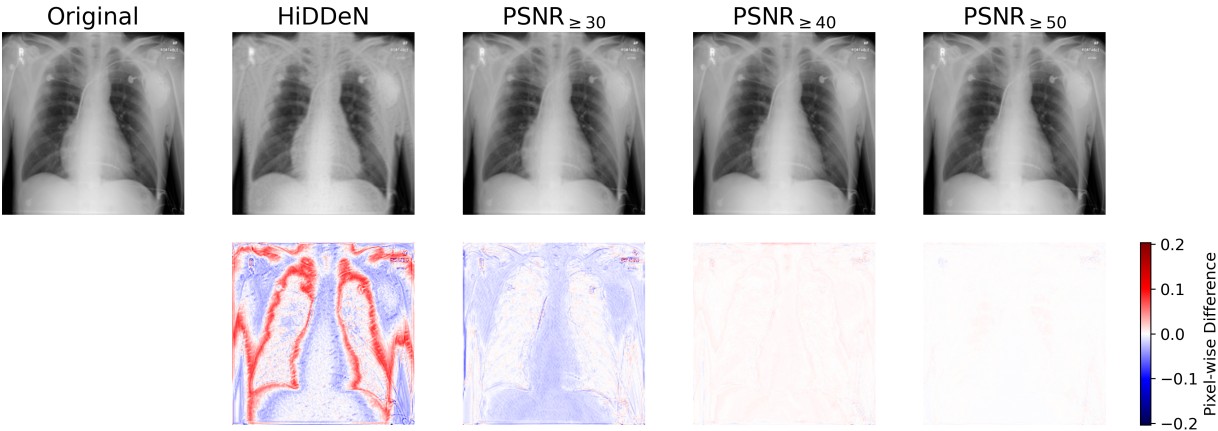

Figure 7: Visual comparison of original and watermarked images for the four model configurations. For each watermarked image, a corresponding heatmap illustrates the pixel-wise differences with respect to the original image.

drop for higher thresholds. These results indicate that the proposed constraint enables the model to produce watermarked images that preserve the diagnostic integrity of the originals.

Table 3: Per-pathology AUROC. DenseNet-121 baseline results and signed differences relative to the baseline for the four watermarking configurations.

| | Base Classifier | HiDDeN | PSNR$_{\geq 30}$ | PSNR$_{\geq 40}$ | PSNR$_{\geq 50}$ |
|---|---|---|---|---|---|
| Atelectasis | 0.771 | $-0.077$ | $-0.018$ | $0.000$ | $+0.001$ |
| Cardiomegaly | 0.883 | $-0.064$ | $-0.013$ | $-0.001$ | $0.000$ |
| Effusion | 0.832 | $-0.054$ | $-0.009$ | $-0.001$ | $-0.001$ |
| Infiltration | 0.710 | $-0.038$ | $-0.001$ | $0.000$ | $0.000$ |
| Mass | 0.813 | $-0.114$ | $-0.021$ | $-0.001$ | $0.000$ |
| Nodule | 0.764 | $-0.091$ | $-0.023$ | $+0.001$ | $0.000$ |
| Pneumonia | 0.711 | $-0.054$ | $-0.010$ | $-0.001$ | $0.000$ |
| Pneumothorax | 0.881 | $-0.158$ | $-0.015$ | $-0.001$ | $0.000$ |
| Consolidation | 0.749 | $-0.080$ | $-0.015$ | $0.000$ | $0.000$ |
| Edema | 0.850 | $-0.036$ | $-0.004$ | $0.000$ | $-0.001$ |
| Emphysema | 0.910 | $-0.200$ | $-0.016$ | $-0.002$ | $0.000$ |
| Fibrosis | 0.840 | $-0.090$ | $-0.010$ | $+0.001$ | $0.000$ |
| Pleural_Thickening | 0.781 | $-0.066$ | $-0.009$ | $-0.001$ | $0.000$ |
| Hernia | 0.858 | $-0.102$ | $-0.021$ | $+0.002$ | $+0.001$ |

## 6 Conclusions

In this paper, we proposed and proved convergence results for a (stochastic) sequential penalty method tailored for learning problems where part of the requirements appear in the form of constraints of the underlying optimization problem. The approach is tested on image processing task, with particular emphasis on a watermarking application. The results show that the methodology can be successfully employed to handle these scenarios. Future research might focus, on the one hand, in the exploitation of the proposed method in other image processing applications. On the other hand, the proposed algorithmic approach could be further refined in order to make it more robust and adaptive to the initial choice of the penalty parameter and its increase rate. Moreover, a tailored analysis for the case of the $\ell_1$ and other nonsmooth penalty functions would surely be of interest, as well as finite-sample complexity studies.

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

## A  Proof of the Finite Termination of the Inner Solver

In this section we report the proofs of Lemma 3 and of Theorem 2.

**Lemma 3.** *Let $C \subseteq \mathbb{R}^n$ be a convex compact set. The penalty function $P_\tau$ associated with problem (2) is $L_{\tau,C}$-smooth with $L_{\tau,C} = L_f + \tau\big(\sum_{i=1}^m M_{i1}^2 + M_{i2}L_g\big)$, where*

$$M_{i1} := \sup_{x \in C} \|\nabla g_i(x)\|, \qquad M_{i2} := \sup_{x \in C} \max(0, g_i(x)).$$

*Proof.* $P_\tau$ is a differentiable function with

$$\nabla P_\tau(x) = \nabla f(x) + \tau \sum_{i=1}^m \max\{0, g_i(x)\}\nabla g_i(x),$$

which can be easily shown to be continuous.

Now, for the ease of notation let $g_i^+(x) = \max\{0, g_i(x)\}$. Let $x, y \in C$. We have

$$\|\nabla P_\tau(x) - \nabla P_\tau(y)\| \leq \|\nabla f(x) - \nabla f(y)\| + \tau \left\|\sum_{i=1}^m g_i^+(x)\nabla g_i(x) - g_i^+(y)\nabla g_i(y)\right\|$$

$$\leq \|\nabla f(x) - \nabla f(y)\| + \tau \sum_{i=1}^m \|g_i^+(x)\nabla g_i(x) - g_i^+(y)\nabla g_i(y)\|.$$

We can now rearrange the terms in the sums, adding and subtracting $\max\{0, g_i(y)\}\nabla g_i(x)$, to get

$$g_i^+(x)\nabla g_i(x) - g_i^+(y)\nabla g_i(y) = (g_i^+(x) - g_i^+(y))\nabla g_i(x) + g_i^+(y)(\nabla g_i(x) - \nabla g_i(y)).$$

Taking norms, using triangle inequality, and recalling the definitions of $M_{i1}$ and $M_{i2}$, we get

$$\|g_i^+(x)\nabla g_i(x) - g_i^+(y)\nabla g_i(y)\| \leq |g_i^+(x) - g_i^+(y)|\|\nabla g_i(x)\| + g_i^+(y)\|\nabla g_i(x) - \nabla g_i(y)\|$$
$$\leq |g_i^+(x) - g_i^+(y)|M_{i1} + \|\nabla g_i(x) - \nabla g_i(y)\|M_{i2}.$$

From the properties of the max function, we have that $|g_i^+(x) - g_i^+(y)| \leq |g_i(x) - g_i(y)|$. Then, from the mean value theorem it holds that $|g_i(x) - g_i(y)| = \|\nabla g_i(z)\|\|x - y\| \leq M_{i1}\|x - y\|$, where the second equality follows since $z$ lies in the line segment connecting $x$ and $y$, and therefore $z \in C$. Recalling that $g_i$ is $L_{g_i}$-smooth, we can continue writing

$$\|g_i^+(x)\nabla g_i(x) - g_i^+(y)\nabla g_i(y)\| \leq |g_i^+(x) - g_i^+(y)|M_{i1} + \|\nabla g_i(x) - \nabla g_i(y)\|M_{i2}$$
$$\leq M_{i1}^2\|x - y\| + M_{i2}L_{g_i}\|x - y\|.$$

Putting everything back together, recalling that $f$ is $L_f$-smooth, we get

$$\|\nabla P_\tau(x) - \nabla P_\tau(y)\| \leq (L_f + \tau(\sum_{i=1}^m M_{i1}^2 + M_{i2}L_{g_i}))\|x - y\|,$$

which completes the proof.

$\square$

**Theorem 2.** *Let $\{z^t\}$ be the sequence produced by SGD, with a constant stepsize $\eta = \frac{1}{\rho_{\tau_k}L_{\tau_k,C}}$ applied to problem (5), i.e., by updates of the form $z^{t+1} = z^t - \eta\nabla P_{\tau_k}^{i_t}(z^t)$, assuming that $P_{\tau_k}$ satisfies the SGC property with an SGC constant $\rho_{\tau_k}$. Further assume that there exists a convex compact set $C \subseteq \mathbb{R}^n$ such that $\{z^t\} \subseteq C$ and that, at each iteration $t$, the algorithm outputs a solution $\hat{x}^t$ uniformly drawn from $\{z^0, \ldots, z^{t-1}\}$, i.e., $\hat{x}^t \sim \mathcal{U}[z^0, \ldots, z^{t-1}]$. Then, for any $\epsilon_k > 0$, we have*

$$\mathbb{E}[\|\nabla P_{\tau_k}(\hat{x}^t)\|] \leq \epsilon_k$$

*for all $t \geq T_k$, with $T_k = \frac{2\rho_{\tau_k}L_{\tau_k,C}(P_{\tau_k}(z^0) - P_{\tau_k}^*)}{\epsilon_k^2}$.*

*Proof.* Note that, since $f(z) \geq f^*$ for all $z \in \mathbb{R}^n$ and that $P_{\tau_k}(z) \geq f(z)$ for all $z$ by the definition of $P_{\tau_k}$, we have that the finite value $f^*$ represents a lower bound for $P_{\tau_k}$. So, $P_{\tau_k}^* = \inf_{x \in \mathbb{R}^n} P_{\tau_k}(x)$ is finite.

Now, by Lemma 3, it holds that $P_{\tau_k}$ is $L_{\tau_k,C}$-smooth on $C$, and since $\{z^t\} \subseteq C$ we can therefore apply the descent lemma (Grippo & Sciandrone, 2023, Prop. 11.3) and write for all $t$

$$P_{\tau_k}(z^{t+1}) \leq P_{\tau_k}(z^t) + \nabla P_{\tau_k}(z^t)^\top(z^{t+1} - z^t) + \frac{L_{\tau_k,C}}{2}\|z^{t+1} - z^t\|^2$$

$$= P_{\tau_k}(z^t) - \eta\nabla P_{\tau_k}(z^t)^\top\nabla P_{\tau_k}^{i_t}(z^t) + \frac{\eta^2 L_{\tau_k,C}}{2}\|\nabla P_{\tau_k}^{i_t}(z^t)\|^2$$

and, rearranging,

$$\frac{P_{\tau_k}(z^{t+1}) - P_{\tau_k}(z^t)}{\eta} \leq -\nabla P_{\tau_k}(z^t)^\top\nabla P_{\tau_k}^{i_t}(z^t) + \frac{\eta L_{\tau_k,C}}{2}\|\nabla P_{\tau_k}^{i_t}(z^t)\|^2.$$

Taking the expectation conditioned to $z^t$, from the assumption of unbiased gradient estimates, it holds

$$\mathbb{E}_{i_t}\left[\frac{P_{\tau_k}(z^{t+1}) - P_{\tau_k}(z^t)}{\eta}\right] \leq -\|\nabla P_{\tau_k}(z^t)\|^2 + \frac{\eta L_{\tau_k,C}}{2}\mathbb{E}_{i_t}\left[\|\nabla P_{\tau_k}^{i_t}(z^t)\|^2\right];$$

then, since $P_{\tau_k}$ satisfies the SGC, we get

$$\mathbb{E}_{i_t}\left[\frac{P_{\tau_k}(z^{t+1}) - P_{\tau_k}(z^t)}{\eta}\right] \leq -\|\nabla P_{\tau_k}(z^t)\|^2 + \frac{\rho_{\tau_k}\eta L_{\tau_k,C}}{2}\|\nabla P_{\tau_k}(z^t)\|^2.$$

Rearranging the terms, we get

$$\left(1 - \frac{\rho_{\tau_k}\eta L_{\tau_k,C}}{2}\right)\|\nabla P_{\tau_k}(z^t)\|^2 \leq \mathbb{E}_{i_t}\left[\frac{P_{\tau_k}(z^t) - P_{\tau_k}(z^{t+1})}{\eta}\right],$$

or equivalently, from the definition of $\eta$,

$$\frac{1}{2}\|\nabla P_{\tau_k}(z^t)\|^2 \leq \rho_{\tau_k}L_{\tau_k,C}\mathbb{E}_{i_t}\left[P_{\tau_k}(z^t) - P_{\tau_k}(z^{t+1})\right].$$

Taking the total expectation we obtain

$$\mathbb{E}\left[\|\nabla P_{\tau_k}(z^t)\|^2\right] \leq 2\rho_{\tau_k}L_{\tau_k,C}\mathbb{E}\left[P_{\tau_k}(z^t) - P_{\tau_k}(z^{t+1})\right].$$

Then, summing over $T$ iterations we get

$$\sum_{t=0}^{T-1}\mathbb{E}\left[\|\nabla P_{\tau_k}(z^t)\|^2\right] \leq 2\rho_{\tau_k}L_{\tau_k,C}\sum_{t=0}^{T-1}\mathbb{E}\left[P_{\tau_k}(z^t) - P_{\tau_k}(z^{t+1})\right]$$
$$= 2\rho_{\tau_k}L_{\tau_k,C}\mathbb{E}\left[P_{\tau_k}(z^0) - P_{\tau_k}(z^T)\right],$$

from which, using that $P_{\tau_k}(z^T) \geq P_{\tau_k}^*$, we get

$$\sum_{t=0}^{T-1}\mathbb{E}[\|\nabla P_{\tau_k}(z^t)\|^2] \leq 2\rho_{\tau_k}L_{\tau_k,C}(P_{\tau_k}(z^0) - P_{\tau_k}^*).$$

Dividing both sides by $T$ we then get

$$\frac{1}{T}\sum_{t=0}^{T-1}\mathbb{E}[\|\nabla P_{\tau_k}(z^t)\|^2] \leq \frac{1}{T}2\rho_{\tau_k}L_{\tau_k,C}(P_{\tau_k}(z^0) - P_{\tau_k}^*).$$

Now, since $\hat{x}^T$ is uniformly sampled from $\{z^0, \ldots, z^{T-1}\}$, we have that the leftmost expression in the above inequality represents the expected value of $\|\nabla P_{\tau_k}(\hat{x}^T)\|^2$. We can then write:

$$\mathbb{E}[\mathbb{E}[\|\nabla P_{\tau_k}(\hat{x}^T)\|^2]] = \mathbb{E}[\|\nabla P_{\tau_k}(\hat{x}^T)\|^2] \leq \frac{2\rho_{\tau_k}L_{\tau_k,C}(P_{\tau_k}(z^0) - P_{\tau_k}^*)}{T}.$$

Also, by Jensen's inequality we can write

$$\mathbb{E}[\|\nabla P_{\tau_k}(\hat{x}^T)\|^2] \geq \mathbb{E}[\|\nabla P_{\tau_k}(\hat{x}^T)\|]^2.$$

We finally get $\mathbb{E}[\|\nabla P_{\tau_k}(\hat{x}^T)\|] \leq \epsilon_k$ if

$$\sqrt{\frac{2\rho_{\tau_k}L_{\tau_k,C}(P_{\tau_k}(z^0) - P_{\tau_k}^*)}{T}} \leq \epsilon_k,$$

i.e., $T \geq \frac{2\rho_{\tau_k}L_{\tau_k,C}(P_{\tau_k}(z^0) - P_{\tau_k}^*)}{\epsilon_k^2}$. $\qquad\square$

# B  Proof of the Outer Loop Convergence

In this section we report the proof of Theorem 3.

**Theorem 3.** *Consider problem (2) and let $P_\tau$ be the associated penalty function. Assume $C \subseteq \mathbb{R}^n$ is a compact set and $\{x^k\} \subseteq C$ is such that*

$$\mathbb{E}\big[\|\nabla P_{\tau_k}(x^k)\|\big] \le \epsilon_k,$$

*for two sequences $\{\tau_k\}$ and $\{\epsilon_k\}$ such that $\tau_k \to \infty$ and $\epsilon_k \to 0$. Then $\|\nabla P_{\tau_k}(x^k)\| \xrightarrow{P} 0$ and there exists a subsequence of indices $\{k_j\}$ such that $\|\nabla P_{\tau_{k_j}}(x^{k_j})\| \xrightarrow{a.s.} 0$. Moreover, almost surely there exists a limit point $\bar{x}$ of $\{x^k\}$ such that, if it satisfies the E-LICQ, then it is a feasible solution for problem (2), i.e., $\bar{x} \in S$, and it is a KKT point for the original problem.*

*Proof.* Let $\eta > 0$. Recalling Lemma 2 and the assumption on sequence $\{x^k\}$ we can write

$$\mathbb{P}\big(\|\nabla P_{\tau_k}(x^k)\| > \eta\big) \le \frac{\mathbb{E}[\|\nabla P_{\tau_k}(x^k)\|]}{\eta} \le \frac{\epsilon_k}{\eta}.$$

Since $\epsilon_k \to 0$ we immediately get that

$$\lim_{k \to \infty} \mathbb{P}\big(\|\nabla P_{\tau_k}(x^k)\| > \eta\big) = 0.$$

Since $\eta$ is an arbitrary positive value, we can conclude that $\|\nabla P_{\tau_k}(x^k)\| \xrightarrow{P} 0$. Then, by Lemma 1, there exists a subsequence $\{k_j\} \subseteq \{1, 2, \dots\}$ such that $\|\nabla P_{\tau_{k_j}}(x^{k_j})\| \xrightarrow{a.s.} 0$.

Now, let $\omega$ be any event from the probability-1 set where the limit holds, so that $\{x^{k_j}(\omega)\}$ is a sample-path such that $\nabla P_{\tau_{k_j}}(x^{k_j}(\omega)) \to 0$. Since $\{x^{k_j}(\omega)\}$ is contained within the compact set $C$, it admits a convergent subsequence (still denoted $\{x^{k_j}(\omega)\}$ for simplicity) such that $x^{k_j}(\omega) \to \bar{x}(\omega)$. We assume that the E-LICQ holds at $\bar{x}(\omega)$.

By the definition of $P_{\tau_k}$, we know that for $k_j \to \infty$, it holds

$$\|\nabla f(x^{k_j}(\omega)) + \tau_{k_j} \sum_{i=1}^{m} \max\{0, g_i(x^{k_j}(\omega))\} \nabla g_i(x^{k_j}(\omega))\| \to 0.$$

Recalling that $\tau_k \to \infty$, we can also observe that

$$\frac{1}{\tau_{k_j}} \|\nabla f(x^{k_j}(\omega)) + \tau_{k_j} \sum_{i=1}^{m} \max\{0, g_i(x^{k_j}(\omega))\} \nabla g_i(x^{k_j}(\omega))\| \to 0.$$

Since $\nabla f$, $\nabla g_i$s are continuous, in the limit along the convergent subsequence we get $\|\sum_{i=1}^{m} \max\{0, g_i(\bar{x}(\omega))\} \nabla g_i(\bar{x}(\omega))\| = 0$, i.e.,

$$\sum_{i=1}^{m} \max\{0, g_i(\bar{x}(\omega))\} \nabla g_i(\bar{x}(\omega)) = \sum_{i \in I_+(\bar{x}(\omega))} \max\{0, g_i(\bar{x}(\omega))\} \nabla g_i(\bar{x}(\omega)) = 0.$$

By the E-LICQ, we know that vectors $\nabla g_i(\bar{x}(\omega))$, $i \in I_+(\bar{x}(\omega))$, are linearly independent, and thus $\max\{0, g_i(\bar{x}(\omega))\} = 0$ for all $i \in I_+(\bar{x}(\omega))$, i.e., there is no $i \in \{1, \dots, m\}$ such that $g_i(\bar{x}(\omega)) > 0$. Hence $\bar{x}(\omega) \in S$.

Now, let us go back to

$$\|\nabla f(x^{k_j}(\omega)) + \tau_{k_j} \sum_{i=1}^{m} \max\{0, g_i(x^{k_j}(\omega))\} \nabla g_i(x^{k_j}(\omega))\| \to 0,$$

and let, for every $k_j$ in the subsequence and every $i$, $\lambda_i^{k_j}(\omega) = \tau_{k_j} \max\{0, g_i(x^{k_j}(\omega))\}$. The sequence $\{\lambda^{k_j}\}$ is bounded. In fact, assume by contradiction that $\|\lambda^{k_j}(\omega)\| \to \infty$ and let us define $\bar{\lambda}^{k_j}(\omega) = \lambda^{k_j}(\omega)/\|\lambda^{k_j}(\omega)\|$. The sequence $\{\bar{\lambda}^{k_j}(\omega)\}$ is bounded by definition, as $\|\bar{\lambda}^{k_j}(\omega)\| = 1$ for all $k_j$. Dividing the argument of the above limit by $\|\lambda^{k_j}(\omega)\|$ and taking the limits, along a further subsequence where $\bar{\lambda}^{k_j}(\omega) \to \bar{\lambda}(\omega)$ if needed, recalling $\|\lambda^{k_j}(\omega)\| \to \infty$ and the continuity of $\nabla f$ and $\nabla g_i$s, we get $\|\sum_{i=1}^m \bar{\lambda}_i(\omega)\nabla g_i(\bar{x}(\omega))\| = 0$, i.e.,

$$\sum_{i=1}^m \bar{\lambda}_i(\omega)\nabla g_i(\bar{x}(\omega)) = 0.$$

We shall note that $\lambda_i^{k_j}(\omega) \geq 0$ by definition for all $i$ and $k_j$; $\bar{\lambda}^{k_j}(\omega)$ are then also all nonnegative. Hence, in the limit we have $\bar{\lambda}_i(\omega) \geq 0$ for all $i$. Moreover, for all $i \notin I(\bar{x}(\omega))$ we will have $\bar{\lambda}_i^{k_j}(\omega) = 0$ for all $k_j$ sufficiently large, so that $\bar{\lambda}_i(\omega) = 0$ for all $i \notin I(\bar{x}(\omega))$. But then

$$\sum_{i \in I(\bar{x}(\omega))} \bar{\lambda}_i(\omega)\nabla g_i(\bar{x}(\omega)) = 0,$$

which by the LICQ is only possible if $\bar{\lambda}_i(\omega) = 0$ for all $i \in I(\bar{x}(\omega))$, but then $\bar{\lambda}(\omega) = 0$, which is absurd since it is the limit of a sequence of unit vectors.

Hence, we have $\{\lambda^{k_j}(\omega)\}$ is a bounded sequence. Taking the limits in

$$\|\nabla f(x^{k_j}(\omega)) + \sum_{i=1}^m \lambda^{k_j}(\omega)\nabla g_i(x^{k_j}(\omega))\| \leq \epsilon_{k_j},$$

along a further subsequence if needed where $\lambda^{k_j}(\omega) \to \bar{\lambda}(\omega)$, recalling the continuity of of $\nabla f$ and $\nabla g_i$s, we get

$$\nabla f(\bar{x}(\omega)) + \sum_{i=1}^m \bar{\lambda}_i(\omega)\nabla g_i(\bar{x}(\omega)) = 0, \tag{6}$$

with $\bar{\lambda}(\omega) \geq 0$ by definition and $g(\bar{x}(\omega)) \leq 0$ by the feasibility result proven above. We can also note that $\lambda_i^{k_j}(\omega) = 0$ for all $i \notin I(\bar{x}(\omega))$ for all $k_j$ sufficiently large, and thus $\bar{\lambda}_i(\omega) = 0$ for all $i \notin I(\bar{x}(\omega))$. We therefore have

$$\bar{\lambda}_i(\omega)g_i(\bar{x}(\omega)) = 0 \text{ for all } i. \tag{7}$$

Putting together feasibility of $\bar{x}(\omega)$, $\bar{\lambda}(\omega) \geq 0$, equation 6 and equation 7, we can conclude that the accumulation point $\bar{x}(\omega)$ is a KKT point of the original problem.

Since an event $\omega$ such that $\nabla P_{\tau_k}(x^{k_j}) \to 0$ occurs almost surely, the proof is thus complete. $\qquad \square$

## C    Results using the Quadratic Penalty Method

In this section we report the results of using a quadratic penalty to solve the constrained MNIST reconstruction and classification problem described in Section 5.1. The experimental setting mirrors that used in Section 5.1.

We report in Tables 4 and 5 the train and test performances of the models obtained using the sequential quadratic penalty method, and compare them with those of the models trained using a linear penalty, previously reported in Tables 1 and 2 of the main paper. We observe that the quadratic penalty struggles to obtain a large number of satisfied constraints, even for large values of the initial penalty parameter. We note that the linear penalty is substantially more effective at obtaining a large number of satisfied constraints, while maintaining high classification accuracy. In fact, in all our tests with the quadratic penalty we are always able to find a configuration where the linear penalties achieved both better classification accuracy and a larger number of satisfied constraints, with the only exception being the case with $\tau_0 = 100$, where the quadratic penalty is not working effectively and is substantially ignoring the constraints.

Table 4: Train performances of the sequential quadratic penalty method in the constrained MNIST reconstruction and classification problem, together with the corresponding results obtained using the linear penalty. Average values over three runs are reported, together with the standard deviation.

| [TRAIN] | $\tau_0$ | $\gamma$ | $\ell_{\mathrm{CE}}$ ($\times 10^{-2}$) | Accuracy (%) | $\ell_{\mathrm{MSE}}$ ($\times 10^{-3}$) | Violation ($\times 10^{-4}$) | Satisfied (%) |
|---|---|---|---|---|---|---|---|
| Lin. Penalty | 10 | 1.01 | $5.631 \pm 0.120$ | $98.42 \pm 0.06$ | $10.865 \pm 0.031$ | $2.200 \pm 0.012$ | $51.93 \pm 0.39$ |
| Lin. Penalty | 50 | 1.01 | $8.404 \pm 0.048$ | $97.54 \pm 0.03$ | $8.650 \pm 0.030$ | $0.829 \pm 0.004$ | $72.73 \pm 0.03$ |
| Lin. Penalty | 100 | 1.01 | $9.934 \pm 0.038$ | $97.09 \pm 0.05$ | $8.206 \pm 0.059$ | $0.600 \pm 0.020$ | $77.68 \pm 0.31$ |
| Quadr. Penalty | 100 | 1.01 | $4.739 \pm 0.735$ | $98.66 \pm 0.31$ | $16.672 \pm 0.185$ | $6.839 \pm 0.169$ | $11.49 \pm 0.77$ |
| Quadr. Penalty | 1000 | 1.01 | $5.729 \pm 0.285$ | $98.40 \pm 0.15$ | $11.255 \pm 0.035$ | $2.213 \pm 0.043$ | $44.15 \pm 0.58$ |
| Quadr. Penalty | 1000 | 1.02 | $10.442 \pm 0.162$ | $96.90 \pm 0.06$ | $9.209 \pm 0.074$ | $0.942 \pm 0.010$ | $63.10 \pm 0.53$ |
| Quadr. Penalty | 10000 | 1.01 | $9.542 \pm 0.170$ | $97.22 \pm 0.05$ | $9.084 \pm 0.072$ | $0.845 \pm 0.039$ | $64.61 \pm 0.79$ |

Table 5: Test performances of the sequential quadratic penalty method in the constrained MNIST reconstruction and classification problem, together with the corresponding results obtained using the linear penalty. Average values over three runs are reported, together with the standard deviation.

| [TEST] | $\tau_0$ | $\gamma$ | $\ell_{\mathrm{CE}}$ ($\times 10^{-2}$) | Accuracy (%) | $\ell_{\mathrm{MSE}}$ ($\times 10^{-3}$) | Violation ($\times 10^{-4}$) | Satisfied (%) |
|---|---|---|---|---|---|---|---|
| Lin. Penalty | 10 | 1.01 | $8.878 \pm 0.346$ | $97.33 \pm 0.13$ | $10.748 \pm 0.022$ | $2.181 \pm 0.003$ | $53.38 \pm 0.16$ |
| Lin. Penalty | 50 | 1.01 | $10.794 \pm 0.162$ | $96.72 \pm 0.05$ | $8.813 \pm 0.029$ | $1.073 \pm 0.007$ | $70.30 \pm 0.26$ |
| Lin. Penalty | 100 | 1.01 | $11.765 \pm 0.435$ | $96.53 \pm 0.03$ | $8.463 \pm 0.053$ | $0.923 \pm 0.021$ | $73.78 \pm 0.14$ |
| Quadr. Penalty | 100 | 1.01 | $8.632 \pm 0.955$ | $97.51 \pm 0.25$ | $16.289 \pm 0.197$ | $6.481 \pm 0.177$ | $13.00 \pm 0.94$ |
| Quadr. Penalty | 1000 | 1.01 | $8.935 \pm 0.281$ | $97.26 \pm 0.09$ | $11.193 \pm 0.018$ | $2.248 \pm 0.027$ | $46.62 \pm 0.21$ |
| Quadr. Penalty | 1000 | 1.02 | $12.124 \pm 0.500$ | $96.40 \pm 0.20$ | $9.461 \pm 0.090$ | $1.289 \pm 0.023$ | $63.85 \pm 0.83$ |
| Quadr. Penalty | 10000 | 1.01 | $11.464 \pm 0.417$ | $96.57 \pm 0.15$ | $9.353 \pm 0.077$ | $1.203 \pm 0.040$ | $65.16 \pm 0.71$ |

We analyze the distribution of the reconstruction errors $\ell_{\mathrm{MSE}}(I_j, \hat{I}_j)$ over the training and test sets to understand why quadratic penalties are ineffective at strictly enforcing the constraints. As we report in Figure 8, we observe that even when the penalty parameter is large, many samples exhibit a very small violation, however, they still fail to satisfy the required threshold. This effect is a direct consequence of the employment of the $\ell_2$ norm, which causes gradients in the penalty functions to become very small when the violation is close to zero. On the contrary, the linear penalty is capable of bringing to zero violation a large number of samples, thanks to the well-known sparsifying behavior of the $\ell_1$ norm.

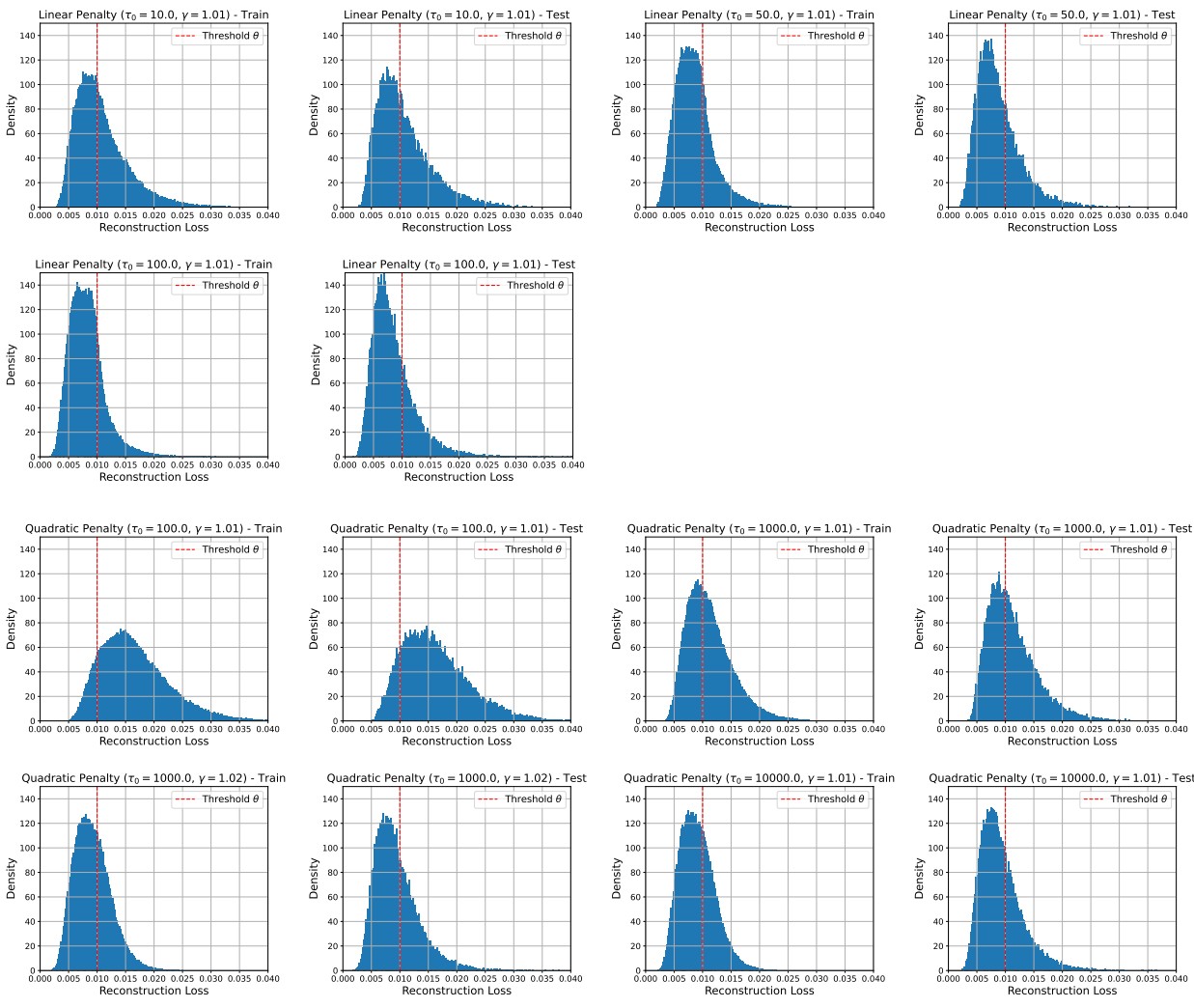

Figure 8: Train and test densities of the reconstruction loss for the sequential quadratic penalty method (bottom) and the sequential linear penalty method (top) in the constrained MNIST reconstruction and classification problem. Density distributions are obtained combining the reconstruction losses of three independent runs.

# D    Additional results on the Medical Image Watermarking problem

In this section, we report the distribution of the PSNR $(I_{co}, I_{en})$ over the training and test sets of the image watermarking problem considered in Section 5.2. As we observe in Figure 9 the sequential penalty method is effectively enforcing the constraints on all training samples as long as the required threshold is not too high. On the other hand HiDDeN remains centered around a PSNR of approximately 30 on both training and test data.

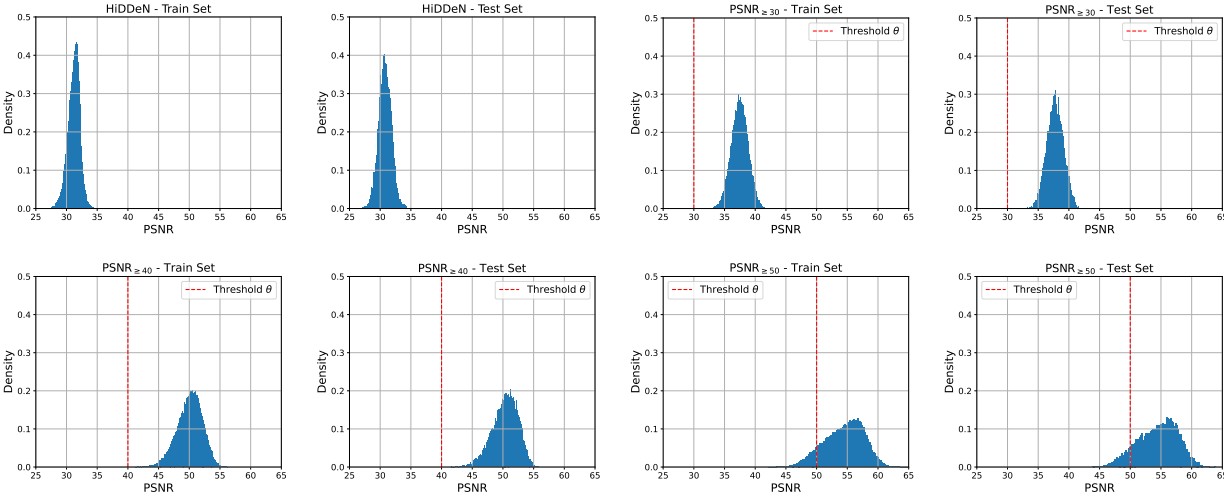

Figure 9: Train and test densities of the PSNR for HiDDeN and for the sequential penalty method in the medical image watermarking problem.

# E    Algorithm Pseudo-Code

---

**Algorithm 1** Convergent Penalty Framework

---

**Require:** $x^0 \in \mathbb{R}^n$, $f : \mathbb{R}^n \to \mathbb{R}$, $g_i : \mathbb{R}^n \to \mathbb{R} \ \forall i$, $\{\tau_k\}$ such that $\tau_k \to \infty$, $\{\epsilon_k\}$ such that $\epsilon_k \to 0$

1: **for** $k = 0, 1, \dots$ **do**

2:    Define

$$P_{\tau_k}(x) = f(x) + \frac{\tau_k}{2} \sum_{i=1}^{m} \max\{0, g_i(x)\}^2$$

3:    Run a (stochastic) optimization procedure, that produces a point $\hat{x}^k$ with the guarantee that that

$$\mathbb{E}[\|\nabla P_{\tau_k}(\hat{x}^k)\|] \leq \epsilon_k$$

4:    Update

$$x^{k+1} \leftarrow \hat{x}^k$$

5: **end for**

---

**Algorithm 2** Practical Penalty Framework

---

**Require:** $x^0 \in \mathbb{R}^n$, $f_j : \mathbb{R}^n \to \mathbb{R} \ \forall j$, $g_{ij} : \mathbb{R}^n \to \mathbb{R} \ \forall i, \ \forall j$, $\tau_0 > 0$, $\gamma > 1$, $\eta > 0$, $\{T_k\}$ such that $T_k \in \mathbb{N}_{>0}$

1: **for** $k = 0, 1, \dots$ **do**

2:    Define

$$P_{\tau_k}(x) = \frac{1}{N} \sum_{j=1}^{N} f_j(x) + \frac{\tau_k}{N} \sum_{i=1}^{m} \sum_{j=1}^{N} \max\{0, g_{ij}(x)\} = \frac{1}{N} \sum_{j=1}^{N} P_{\tau_k}^j(x)$$

3:    Let $z^0 \leftarrow x^k$

4:    **for** $t = 0, \dots, T_k - 1$ **do**

5:       Get the mini-batch index set $B_t \subseteq \{1, \dots, N\}$

6:       Evaluate $\nabla P_{\tau_k}^{B_t}(z^t) = \frac{1}{|B_t|} \sum_{j \in B_t} \nabla P_{\tau_k}^j(z^t)$

7:       Compute the SGD-type update direction $d_k$ using $\nabla P_{\tau_k}^{B_t}(z^t)$      ▷ e.g., $d_k$ can be Adam direction

8:       Update $z^{t+1} \leftarrow z^t + \eta d_k$

9:    **end for**

10:    Update

$$x^{k+1} \leftarrow z^{T_k}$$

11:    $\tau_{k+1} = \gamma \tau_k$

12: **end for**

---

