# OpenReview forum: "Sample-wise Constrained Learning via a Sequential Penalty Approach with Applications in Image Processing"
_TMLR — Under review for TMLR_

### Review · Reviewer_uPYm · 2026-05-27

**Summary Of Contributions:**

The paper studies the constrained optimization problem using a sequential penalty method. Some theoretical results are provided that guarantee convergence to the stationary points in expectations. Experiments on image processing tasks are provided to illustrate the efficiency of the algorithm.

**Audience:**

Yes

**Audience Explanation:**

I think so. Constrained optimization and penalty method are always useful and interesting in many machine learning problems.

**Claims And Evidence:**

Yes

**Claims Explanation:**

Mostly yes. But I am uncomfortable with the assumption in Theorem 2 and Theorem 3 that the SGD iterates are always inside the constrained set, without even defining their SGD algorithm properly.

**Requested Changes:**

(1) After equation (3), where here should be where.

(2) Before equation (4), you wrote that you have to settle for a result of the type $\mathbb{E}[\|\nabla P_{\tau_{k}}(x^{k})\|]\leq\epsilon_{k}$. It would be great if you can add more discussions on the context for this type of result is necessary. For example, if the objective is strongly convex, then I expect one can obtain stronger result, i.e. convergence guarantees to global minimizer instead of the stationary point under expectations. The same comment can be said about the discussion in the paragraph after Definition 4. You wrote that fortunately there is no need to exactly solve each subproblem to global optimality. Please add more discussions. It is not clear to me why there is no need to find global optimality. I think finding global optimality might be desirable in some settings. If in general it is impossible to find global optimality, instead of no need to find global optimality, you need to rephrase your discussions.

(3) It would be great if you can add some discussions about for which particular applications in machine learning the strong growth condition in Definition 5 is satisfied after Definition 5.

(4) The statement of Lemma 3 is a bit confusing to me. You introduced the convex compact set $C$ here, which was never introduced before. Is $C$ representing the constraint in equation (2)? Please make it more clear. Another thing that confuses me is that in the statement of Lemma 3, you are talking about problem (2), but right before it, you stated problem (5), and later in Theorem 2, you are also talking about problem (5). If Lemma 3 intends for problem (2), perhaps there is no need to introduce problem (5) until before Theorem 2.
Later, in Theorem 3, you talk about problem (2) again, instead of (5). This causes another confusion.

(5) Even though SGD is a well-known algorithm, sometimes it takes various forms under various settings and assumptions.
To be more transparent, I recommend you to define the SGD iterates $z^t$ explicitly before you start Theorem 2.

(6) My understanding is that for the penalty method, it is a soft constraint, i.e. with high probability, the SGD iterates will not get outside the constrained set $C$. As a result, it is not clear to me how you can guarantee that $z^{t}\in C$ in Theorem 2. It seems that right now you simply assumed that as in the statement of Theorem 2. I think that statement is too strong because in the results for Theorem 2, it is about $\mathbb{E}[\|\nabla P_{\tau_{k}}(\hat{x}^{t})\|]\leq\epsilon_{k}$, which means here you take the expectations, and that means the expectations are taken over all the possibilities, and as I said you cannot assume with probability one that $z^{t}\in C$ for every $t$. The same issue can be said about Theorem 3. My feeling is that in general, such an assumption is definitely too strong because what if the gradient noise in SGD is simply a Gaussian noise? Then, clearly you cannot assume the iterates are inside $C$ at each iteration. However, if the gradient noise is structured in certain way, i.e. in the mini-batching setting or some other settings, maybe that is possible. So it goes back to my previous point that you should also define the SGD algorithm rigorously.

---

> ### Author Response · Authors · 2026-06-22
> **Response to Reviewer uPYm**
>
> We thank the reviewer for their comments, we appreciate the effort of carefully reading our manuscript. We detail here below responses and actions taken to address their concerns and detailed requests.
>
> ## Requested Changes
> ### Request (1)
> We rephrased the sentence, thanks.
>
> ### Request (2)
> We try to clarify our statements. Without convexity assumptions, the obtainment of a globally optimal solution to a (sub)problem is generally not only impractical, but also impossible to guarantee. In this perspective, sequential penalty schemes necessarily had to be devised in such a way that convergence of the overall framework could be achieved by just requiring inexact solutions of the subproblems. This idea is somewhat classical in optimization theory and details can be found for instance in the textbooks of Bertsekas (Nonlinear Programming, 1997) and Grippo & Sciandrone (Introduction to methods for nonlinear optimization, 2023).
>
> Of course, if achieving the global solution of each subproblem is easy, then this can be exploited; however, this is most often the case of convex problems, where the condition $||\nabla f(x_k)||\le \epsilon$ is in fact a direct proxy for (approximate) global optimality.
>
> In order to avoid confusion as underlined by the referee, we rephrased the sentence so that, instead of "there is no need", we now read "it is not necessary".
>
>
> ### Request (3)
> The Strong Growth Condition can be proved to hold under interpolation if the PL-condition holds (Mishkin A., "Interpolation, growth conditions, and stochastic gradient descent", 2020, Lemma 7). In turn, a local version of PL was shown to hold true in large parts of the loss landscape for important deep architectures such as CNNs and ResNets (Liu et al., "Loss landscapes and optimization in over-parameterized non-linear systems and neural networks", 2022).
> In the paper we hence noted this fact when introducing the SGC.
>
> ### Request (4)
> The convex compact set $C$ is not the feasible set defined by the constraints, nor it is related to the constraints in any way. The set is actually introduced as a technical tool for the theoretical analysis.
>
> Basically, the assumption is saying that the entire trajectory of iterates produced by the algorithm remains contained within a set which is possibly enormous, but compact. In other words, we are assuming that the sequence of solutions has no divergent subsequence; if that's the case, the convex hull of the set of iterates is in fact a set satisfying both compactness and convexity.
>
> As said, making this assumption is a theoretical expedient commonly used (see, e.g., Proposition 21.1 in "Introduction to methods for nonlinear optimization", Grippo & Sciandrone, 2023) when analyzing the behavior of sequential penalty approaches, in order to state existence results for accumulation points of the sequences; without the assumption, the analysis could be carried out regardless, and we would obtain results of the form "if the sequence of solutions admits cluster points, then each of these cluster points satisfies...". Here we preferred to follow this setup.
>
>
>
> As for the second part of their comment, we thank the referee for pointing it out, we acknowledge that the previous order of presentation was arguably confusing. In the revised manuscript,  we followed the reviewer's suggestion and switched the order of Lemma 3 and the definition of problem (5).
>
> We finally note that the fact of Theorem 3 holding for any problem of the form (2) was already underlined at the beginning of Section 3.2.
>
>
> ### Request (5)
> We thank the referee for pointing out that the update rule of SGD shall be formalized at some point. We did so in the statement of Theorem 2.
>
>
> ### Request (6)
> With reference also to the response to Request (4), we remark once more that the set $C$ is a standard expedient used for the analysis that allows to also state existence results, together with the characterization of limit points (that are feasible and KKT-stationary). This set can be arbitrarily large and is not related to constraints.

---

### Review · Reviewer_FaeM · 2026-06-01

**Summary Of Contributions:**

This paper studies constrained optimization in machine learning. It proposes to satisfy constraints explicitly instead of adding regularization terms in the loss function. The main algorithm is a sequential method which optimizes under successively stronger quadratic penalties corresponding to the desired constraints; each subproblem is solved approximately by standard SGD-type optimization. This paper proves that this sequential penalty method converges almost surely to a KKT point of the original problem. Performance is then validated via image watermarking experiments on the MNIST and CXR-8 datasets.

Summary of strengths:
1. The mathematics in Section 3 and Section 4 are clearly explained and easy to follow.
2. I found the application to medical image watermarking interesting, particularly the use of PSNR as a principled constraint methodology.

Summary of weaknesses:
1. The claimed advantage of the proposed method is not sufficiently supported.
2. The theoretical and experimental sections study fundamentally different techniques: the theory is done for quadratic penalties while the experiments use linear/absolute penalties.
3. I found the precise contributions of this paper difficult to ascertain, with unclear contextualization in the broader literature.
4. Writing could be improved throughout the paper.

**Additional Comments:**

1. The writing needs significant work throughout the paper: from incorrect grammar (e.g., throughout Section 1) to misspellings (e.g., the name “Kuhn” in Section 2 and “subproblems” in Section 3).
2. The abstract and conclusion are too short and vague for my taste. Similarly, the contributions are not clear from the introduction, and should be specified explicitly.
3. Several references are listed as ArXiv preprints but should instead list the appropriate conference/journal (e.g., the Adam paper was published at ICLR 2015).
4. Definition 1 is incomplete (missing equality to zero for complementary slackness).

**Audience:**

Yes

**Audience Explanation:**

This work is relevant to researchers in constrained optimization, who may find the theoretical results worthwhile, as well as the interpretability community as a theoretically-justified method for enforcing semantic constraints in machine learning.

On the other hand, I found it hard to understand the precise contributions of this work and how they relate to the literature. The sequential penalty method itself seems to be well-studied. It’s not clear whether the sample-wise approach (inducing the inner-loop/outer-loop analysis) is a contribution of this paper, or whether this method is common and the main contribution is instead the almost-sure convergence proved in Theorem 3. If the former, how does this decomposition compare to related algorithms in the literature, and what are its advantages and disadvantages? If the latter, why was almost-sure convergence not proven before, and what does this paper propose to do differently? Also, why are asymptotics the main objects of study when finite-sample guarantees would be more applicable to the experimental results?

**Claims And Evidence:**

No

**Claims Explanation:**

While the introduction claims that the advantage of constrained optimization is the removal of the regularization hyperparameter $\lambda$ in the loss function, the proposed sequential penalty method introduces two new hyperparameters: the initial penalty coefficient $\tau_0$ and the multiplicative update scaling $\gamma$. It seems the latter hyperparameters need to be tuned for the best performance (Table 2), negating the claimed advantage. Moreover, proper tuning of $\lambda$ can result in a solution which outperforms the sequential penalty method in terms of satisfied constraints ($\lambda=1000$ in Table 2).

As noted in Section 5, the theoretical and empirical sections of this paper study fundamentally different techniques (quadratic vs linear/absolute penalties, respectively). I believe this greatly weakens the claim that the proposed quadratic sequential penalty method is viable for practical deep learning tasks. The paper states that the quadratic penalty was less stable during training, which is odd as a convergence result was just proven in the previous section -- this suggests a gap between theory and practice which is not sufficiently explained. While I found the medical image watermarking application interesting, particularly the use of the PSNR constraint as an interpretable alternative to image distortion regularization, it is unclear whether the claimed performance in Section 5.2 also applies to the quadratic penalty method.

I checked the proofs and they look right to me (though I am not an expert in constrained optimization theory).

**Requested Changes:**

Critical for acceptance:
1. Clarification of the main claims regarding performance of the sequential penalty method compared to regularization (e.g., number of hyperparameters or Pareto optimality with respect to accuracy and percent of constraints satisfied), and assurance that they are supported by empirical evidence.
2. Unification of the quadratic and linear/absolute penalty methods used throughout the work, i.e., one of the following three approaches: (i) theoretical results for the linear penalty, (ii) empirical results for the quadratic penalty, or (iii) an analysis of the gap between theory and practice for the quadratic penalty which addresses the observed training instability.
3. Specification of the precise contributions of this work, and both a broad and technical comparison to related methods in the literature.

Would strengthen the work:
1. Various improvements to the writing (see Additional Comments).

---

> ### Author Response · Authors · 2026-06-22
> **Response to Reviewer FaeM - Part 1/2**
>
> We greatly thank the referee both for appreciating the core of our work and for their extremely constructive and on-point comments, which, in all honesty, we largely happen to agree with. We answer to their main criticisms here below.
>
> ## Requested Changes
> ### Comparison to regularization approaches and related empirical evidence
> The reviewer points out that, from the perspective of the end user, with the proposed approach there still remains some hyperparameter tuning to obtain the desired performance. We understand the point of view and we acknowledge that it is at least in part true.
>
> However, we would like to point out that the nature of the hyperparameters  that affect the proposed methodology is different w.r.t. the penalized approach; in the former case, the hyperparameters are all related to the algorithm, whereas in the latter case the major hyperparameter is embedded in the learning problem itself. As a consequence, while with penalization the issue of tuning the hyperparameter is intrinsic with the formulation and thus unavoidable, in the case of the constrained formulation there is room for a further development of the algorithmic scheme that could make the approach more robust to the choice of the sequence of penalty parameters.
>
> In addition, we would like to underline that, as we remark in the manuscript, the proposed approach seems to be rather robust to the hyperparameters choice; while some rough tuning is required to achieve the top trade-off, only extreme choices of the parameters lead to massive performance drops; this is not the case for the penalization approach, where a miscalibration of the penalization coefficient has a major impact to the overall performance of the trained model.
>
> ### Specification of the contributions
> We thank the reviewer for making us note that the contributions of this manuscript w.r.t. existing literature could be unclear. To this aim, we added in the Introduction an explicit list of contributions, which are the following ones:
> - we formalize the sample-wise constrained scenario in deep learning, pointing out that relevant problems usually handled by regularization could naturally be cast to this formulation;
> - we propose an algorithmic scheme, easily implementable by  practitioners in standard software frameworks, to deal with the above class of training tasks;
> - we revisit the SGD convergence result for the non-convex case by Vaswani et al. ("Painless stochastic gradient: Interpolation, line-search, and convergence rates", 2019) making explicit the worst case bound on the number of updates needed to drive the expected value of the gradient norm below a given threshold;
> - we provide the (almost sure) convergence analysis, to the best of our knowledge novel in the literature, for the sequential penalty method applied to a general class of problems in the case where the stationarity condition for the subproblems can be obtained only in expectation;
> - we show the viability of the proposed methodology both on a sandbox problem and a real image processing task, offering some insights on the algorithm behavior.

---

> ### Author Response · Authors · 2026-06-22
> **Response to Reviewer FaeM - Part 2/2**
>
> ### Mismatch between theory (l2 penalty) and numerical evidence (l1 penalty)
> We do agree that the mismatch between theory, analyzed for quadratic penalties, and experiments, carried out with the absolute penalty, is unfortunate. From the one hand, however, the theoretical analysis with the $L_1$ penalty (approach (i) suggested by the reviewer) gets much more complicated, for what concerns both the inner and the outer loop, because of the nonsmoothness of the $L_1$ norm. The analysis of SGD without smoothness assumptions and in the nonconvex case is an open research topic, whereas the analysis of (inexact) sequential penalty methods in presence of nonsmooth penalties requires a significant increase in the technical complexity, due to the need of resorting to subdifferential or generalized derivatives for the characterization of the approximate optimality condition of the subproblems. We feel like the analysis in this kind of setup would substantially alter the scope and the positioning of the work, and we would thus prefer not to follow this path.
> On the other hand, we found the performance gap between quadratic and absolute penalty to be substantial - we would have gone for the quadratic penalty in the experiments (approach (ii)) from the start otherwise.
> Instead of cherry-picking some good runs with the quadratic penalty, we thus prefer to follow approach (iii) suggested by the referee, and better highlight the practical limitations and drawbacks of the quadratic penalty and underline the advantage of using the absolute one.
>
> The main issue with the quadratic penalty is that feasibility is satisfied in the limit; however, improvement with constraints violation with a quadratic penalty does not occur linearly with the increase of the penalty parameter: to achieve small violations, it is hence often necessary to drive the penalty parameter to very large values, which in turn make the optimization of the penalty function numerically troublesome. In addition, termination in finite time, even for very large values of $\tau$, quadratic penalties tend to induce very small, but yet nonzero violations across all data points;  this issue is related to gradients of the penalty term that tend to vanish by definition
> when violations are small.
>
> On the contrary, the sparsifying effect of the $L_1$ norm makes the error exactly zero for a large bunch of data points when $\tau$ gets sufficiently large, so that we finally retrieve a solution that is truly satisfying the constraints for most points in the training set.
>
> We now remark explicitly these aspects at the beginning of Section 5. Moreover, we report in a new appendix a numerical comparison between the L1 and L2 penalties on the sandbox problem, in order to support with data the claimed difference in both violation distribution and out-of-sample performance between the two approaches.
>
> Finally, we point out in the revised Conclusions section that the analysis specialized for the case of the absolute penalty would surely be a topic worth of investigation in future research.
>
> ### Finite sample guarantees vs. asymptotic results
> The choice to focus on the asymptotic behavior is in line with most available and consolidated literature on sequential penalty methods and with the fact that this type of analysis is usually the first to be carried out when studying a new algorithmic approach. Finite sample results would certainly be useful, but this would require further efforts that we would prefer to leave for future work. Yet, we point this aspect out in the revised conclusions.
>
>
> ### Additional comments
> #### Writing improvement
> We carefully scanned through the manuscript and did our best to remove all grammar and misprint errors. We thank the referee for pointing this out, we hope that language in the paper is now adequate.
> #### Abstract and conclusions
> We thank the referee for the comment, we have heavily revised the abstract to make it more specific and informative, and the conclusions to include more precise references to possible future developments.
> #### References
> We carefully checked all the references and made all the due fixes. We are grateful to the referee for pointing this out.
> #### Definition 1
> Definition 1 was fixed, thank you.

---

> > ### Comment · Reviewer_FaeM · 2026-06-22
> >
> > Thanks to the authors for the revision and comprehensive response. I especially appreciate the discussion on the quadratic vs linear penalties. I have a couple remaining thoughts/questions on this topic:
> > 1. In Appendix C, could you include one relevant linear penalty result in each table/figure? I found it unwieldy to flip back and forth between the quadratic penalty results in Appendix C and the linear penalty results in Section 5.
> >
> > 2. In your response you state "improvement with constraints violation with a quadratic penalty does not occur linearly with the increase of the penalty parameter". This seems interesting to me -- is this well-known? Why is this statement not included in your introduction to Section 5, even though the rest of the discussion is?
> >
> > 3. The two response paragraphs beginning with "The main issue" and "On the contrary" are, in my opinion, essential for understanding the motivation of Section 5 (and rightly included in the introduction to that section). They are mostly intuitively clear; however, because these insights are a prerequisite for the rest of the section, I believe they need stronger evidence in the text. Are there existing theoretical results you can cite that justify each of these points? Or, even better, could you include a short technical/mathematical justification of these points in the Appendix?

---

> > > ### Author Response · Authors · 2026-06-23
> > > **Response to Reviewer FaeM**
> > >
> > > We thank once again the reviewer for the extremely constructive comments and the very quick feedback.
> > > #### Q1
> > > The referee is right, we have now rearranged Appendix C, so that the results shown in the main text are also reported there now; thanks for pointing it out. For the sake of completeness, we have decided to supplement Figure 8 in Appendix C with four additional density plots for the linear penalty that are not reported in the main paper.
> > > #### Q2
> > > We thank the referee for the question, the underlined issue deserves a clarification. We did not add that particular passage of our response to the paper as it lacked rigor. In fact, the claim is not technically precise: it was intended to provide an intuition, but we certainly needed more care in wording; we apologize about it to the referee.
> > >
> > > The property we had in mind is related to the (sub)gradients of violation terms; in the case of the quadratic penalty, the gradient of the penalty associated with a constraint is proportional to the extent of the violation itself; therefore, as $\tau$ grows and violations get small, the corresponding gradients also get small and further progress towards feasibility becomes slower and slower (this is roughly what we improperly meant by "improvement with constraints violation with a quadratic penalty does not occur linearly with the increase of the penalty parameter").
> > > For the linear penalty, the subgradient of the violations does not contain a factor proportional to the violations themselves, which is why progress in feasibility goes on at a more sustained pace.
> > > #### Q3
> > > We thank the referee for the suggestion, we tried to support the claims at the beginning of Section 5 adding some references. In particular:
> > > - the numerical instability of sequential penalties for large $\tau$ is a well-known issue, discussed for example in Chapter 4.2 of Bertsekas book (similar discussions can be found in other classical manuals on nonlinear optimization); we now explicitly cite this reference in that paragraph;
> > > - the difference in residuals distribution in least-error problems when $\ell_2$ or $\ell_1$ norm are considered is discussed, for example, in Section 1.7 of the book "Numerical Optimization" by Nocedal and Wright. We have added a sentence at the end of the paragraph to refer readers to this reference about this aspect.
> > > - gradients of constraint violations vanishing when violations themselves become small is now underlined by remarking the explicit formula of gradient terms related to the penalty.

---

> > > > ### Comment · Reviewer_FaeM · 2026-06-23
> > > >
> > > > Thanks to the authors for the prompt clarifications. My concerns are satisfied overall.

---

### Review · Reviewer_VhoX · 2026-06-15

**Summary Of Contributions:**

The paper formulates sample-wise requirements in learning as explicit constraints rather than weighted auxiliary losses. It proposes a stochastic sequential-penalty framework for these finite-sum constrained problems, proves subsequence almost-sure convergence to feasible KKT points under classical assumptions, and illustrates the method on MNIST reconstruction/classification and Peak Signal-to-Noise Ratio (PSNR)-constrained medical-image watermarking.

**Audience:**

Yes

**Audience Explanation:**

Yes. The paper addresses a relevant problem -- replacing hard-to-interpret penalty weights with explicit sample-wise constraints in learning problems. This perspective is useful for researchers working on constrained optimization, reliable deep learning, and image-processing applications where per-sample quality thresholds are meaningful.

The theoretical contribution may interest readers concerned with stochastic penalty methods and convergence guarantees, while the medical image watermarking case study may interest practitioners looking for interpretable ways to control image distortion.

**Claims And Evidence:**

Yes

**Claims Explanation:**

The paper gives accurate and reasonably clear evidence for its theoretical claims, namely convergence of a quadratic sequential penalty scheme under compactness, smoothness, strong-growth, and linear independence constraint qualification assumptions.

The evidence is less convincing for the practical claims. The experiments use an absolute/linear penalty and Adam, while the theory analyzes a quadratic penalty with SGD-type assumptions. The authors explicitly acknowledge that the experimental setting is not directly covered by the analysis, but did not offer justification of why these settings are chosen.

The empirical evidence is suggestive but not conclusive. On MNIST, the method improves the tradeoff between accuracy and reconstruction constraint satisfaction, but many sample-wise constraints remain violated at the end of training; e.g. test satisfaction for penalty variants is only about 53 - 75% in Table 2. For watermarking, the PSNR-constrained models improve average PSNR and reduce AUROC degradation relative to the chosen HiDDeN baseline, but the baselines seem limited, and there are no detailed per-sample constraint-satisfaction statistics.

**Requested Changes:**

- To better align theory and experiments, it would be helpful to either add theoretical analysis on the the absolute/linear penalty and Adam-based implementation used in experiments, or add experiments matching the quadratic-penalty SGD setting analyzed theoretically. If neither are possible, please state why explicitly in the paper.
- Would it be possible to report per-sample constraint satisfaction for the watermarking task, including violation rates and distributions, not only average PSNR?

Not critical:
- The experiment section could report multiple random seeds, confidence intervals, and computational cost.
- Explicit pseudocode for both the theoretical algorithm and the practical training procedure might help readers better understand the algorithms.
- In the current writing, it is sometimes unclear how restrictive some  assumptions are, e.g. compactedness, strong growth, E-LICQ.

---

> ### Author Response · Authors · 2026-06-22
> **Response to Reviewer VhoX - Part 1/2**
>
> We thank the referee for the review and, in particular, for the constructive comments. We address their main concerns in the following response.
> ## Requested Changes
>
> ### Mismatch between theory (l2, SGD)  and numerical evidence (l1 penalty, Adam)
> We would split the two questions regarding the mismatch $L_1$-$L_2$ and the mismatch SGD-Adam, as the issues lie on somewhat different levels.
>
> Concerning the choice of the $L_1$ penalty for the experiments, its nonsmooth nature would make the corresponding theoretical analysis quite more complex and we believe it would somewhat change the scope of the paper. Still, the $L_2$ penalty does not provide equally good results from a numerical perspective, in particular concerning the number of constraints satisfied exactly when the algorithm terminates.
> The major difficulty with the quadratic penalty is indeed that feasibility is provably attained asymptotically; however, to achieve small violations, it is often necessary to drive the penalty parameter to very large values, which in turn make the optimization of the penalty function numerically unstable. In addition, even for very large values of $\tau$, quadratic penalties tend to induce very small, but yet nonzero violations across all samples in the training dataset. This issue is related to gradients of the penalty term that tend to vanish by definition when violations are small.
> The $L_1$ penalty on the other hand has a well-known sparsification effect, which makes final solutions fully respect most of the constraints even if the process is terminated "early", and low violations in general are achieved with smaller values of the penalty parameters.
>
> For this reason, in view also of the comment of reviewer FaeM, we described these aspects more in detail at the beginning of Chapter 5 and we also added an experimental comparison between the $L_1$ and $L_2$ penalty on the sandbox problem in a new Appendix to further support these claims with numerical evidence.
>
> As for the choice of Adam over SGD in the experiments: we honestly do not see it a difference as large as the one between the two types of penalties. Considering SGD allows to keep the analysis somewhat to a simpler bar, but a result similar to the one of Vaswani et al. ("Painless stochastic gradient: Interpolation, line-search, and convergence rates", 2019) for SGD is available for Adam (Zhang et al., "Adam can converge without
> any modification on update rules", 2022). The rate of convergence, although comparable, is slightly worse, and requires a diminishing stepsize sequence, but it would be enough to prove both convergence towards stationarity of the inner loop and a bound of the number of steps to achieve in finite time the desired approximate stationarity level in expectation. We now briefly mention this aspect in the revised manuscript, when we describe the experimental setup.
>
>
>
>
> ### Violation analysis in watermarking task
> We thank the referee for this request, the requested plots are surely of interest. The per-sample constraint violation analysis in the watermarking task has been added to the revised manuscript in Appendix D.
>
> ### Other requests
> #### Multiple random seeds
> In the revised manuscript, we have updated all results and figures concerning the first test problem, that now report mean and standard deviations across three independent runs with different random seeds.
>
> We have unfortunately not been able to do the same for the watermarking problem in this short period, because of computing limitations on our side.
> #### Algorithms pseudocodes
> We thank the reviewer for suggesting the insertion of algorithm pseudocode (both theoretical and practical) in the paper. We have placed it in a new Appendix.

---

> ### Author Response · Authors · 2026-06-22
> **Response to Reviewer VhoX - Part 2/2**
>
> #### Restrictiveness of assumptions
> In the revised paper, we have briefly remarked that the SGC should hold true for some relevant learning architectures if we work in the overparametrized regime. More in detail, SGC holds under interpolation + PL condition (Mishkin A., "Interpolation, growth conditions, and stochastic gradient descent", 2020, Lemma 7). A local version of PL was shown to hold true by Liu et al. ("Loss landscapes and optimization in over-parameterized non-linear systems and neural networks", 2022) for architectures like large CNNs or ResNets.
>
> Compactness of $C$ is just a theoretical artifact commonly used to state results of existence of limit points (see, e.g., Proposition 21.1 in "Introduction to methods for nonlinear optimization", Grippo & Sciandrone, 2023), but it does not deal with the constraints of the problem and shall thus not be seen as restrictive.
>
> As for the constraint qualification, we decided to use the E-LICQ for the sake of simplicity of the theoretical analysis. More complicated and less restrictive CQs could have been considered, but additional complexity in the analysis would have been added. We believe such a generalized analysis would arguably go beyond the actual goals and aims of the current manuscript and of the venue we have accordingly chosen.

---

### Author Response · Authors · 2026-06-22
**General response to Reviewers comments**

We thank all the reviewers and the action editor for their work and, in particular, for the constructive comments. We provide answers to every underlined issue, and point out specific actions implemented in the manuscript, in a separate reply to each referee. Here we summarize the major changes that can be found in the revised manuscript w.r.t. the original submission:
- We have rewritten the abstract to make it more detailed and informative about the contents and the results presented in the manuscript;
- We have added a list of contributions at the end of the Introduction;
- We have reorganized the order of presentation of the results in Section 4.1; some algorithmic details have been better specified, also by providing pseudocodes in a new Appendix;
- The gap between theory (devised for the quadratic penalty) and experiments (the linear penalty was used), has been more thoroughly recognized and the reasons for this choice are now discussed with deeper insights; numerical results on this aspect have also been provided in a new Appendix.
- The numerical experiments of Section 5.1 are now presented showing the results obtained with three different random seeds.
- The distribution of constraints violations for the watermarking problem is now reported in a new Appendix, in order to provide more insights about that case study.
- The conclusions now detail future research more broadly.